 **eLiFE**

# Transient inhibition and long-term facilitation of locomotion by phasic optogenetic activation of serotonin neurons

Patrícia A Correia[1†], Eran Lottem[1†], Dhruba Banerjee[1,2], Ana S Machado[1], Megan R Carey[1], Zachary F Mainen[1*]

[1]Champalimaud Neuroscience Program, Champalimaud Centre for the Unknown, Lisbon, Portugal; [2]School of Medicine, University of California, Irvine, United States

**Abstract** Serotonin (5-HT) is associated with mood and motivation but the function of endogenous 5-HT remains controversial. Here, we studied the impact of phasic optogenetic activation of 5-HT neurons in mice over time scales from seconds to weeks. We found that activating dorsal raphe nucleus (DRN) 5-HT neurons induced a strong suppression of spontaneous locomotor behavior in the open field with rapid kinetics (onset $\leq 1$ s). Inhibition of locomotion was independent of measures of anxiety or motor impairment and could be overcome by strong motivational drive. Repetitive place-contingent pairing of activation caused neither place preference nor aversion. However, repeated 15 min daily stimulation caused a persistent increase in spontaneous locomotion to emerge over three weeks. These results show that 5-HT transients have strong and opposing short and long-term effects on motor behavior that appear to arise from effects on the underlying factors that motivate actions.

**\*For correspondence:** zmainen@ neuro.fchampalimaud.org

[†]These authors contributed equally to this work

**Competing interests:** The authors declare that no competing interests exist.

## Introduction

Serotonin (5-HT) is a major neuromodulator that is one of the most important pharmacological targets in the treatment of psychiatric disorders such as anxiety and depression (*Vaswani et al., 2003*). Yet 5-HT has also numerous other clinically-important actions on functions including eating (*Brewerton, 1995*), aggression (*Lesch and Merschdorf, 2000*), sex (*Rosen et al., 1999*), pain (*Jung et al., 1997*), and perception (*Geyer and Vollenweider, 2008*). However, the dominant conception of the 5-HT system still places mood and affect in a central position and considers other actions as side effects. To some degree different behavioral effects have been mapped to specific subsets of the 17-member receptor family, but a coherent pharmacologically-grounded theory of 5-HT function centered on affective states has been elusive (*Dayan and Huys, 2009*).

Optogenetic tools provide novel viewpoints on neuromodulatory function by allowing the neurons that release a neuromodulator to be monitored and manipulated with high anatomical and genetic precision (*Yizhar et al., 2011*). This allows a neuromodulatory system to be studied in terms of firing patterns with a temporal precision unavailable by pharmacological interventions. 5-HT neurons were classically thought to be regular-firing, 'clock-like' neurons (*Jacobs and Azmitia, 1992*). It is now known that, like other neuromodulatory neurons, 5-HT neurons exhibit phasic bursts of firing locked to significant behavioral events, such as stimuli, movements and rewards (*Ranade and Mainen, 2009*; *Cohen et al., 2015*; *Liu et al., 2014*; *Nakamura et al., 2008*). Tonic and phasic activation of the dopamine (*Goto et al., 2007*; *Niv et al., 2007*) and norepinephrine (*Aston-Jones and Cohen, 2005*) systems have distinct functional consequences. Opposite functions for phasic and

**eLife digest** The brain controls sleep, movement and the other behaviors that an animal needs to survive. A chemical called serotonin plays an important role in controlling these behaviors as it regulates the activity of nerve cells (known as neurons) throughout the brain. Serotonin is produced by a specific group of neurons found in an area at the base of the brain called the raphe nuclei. From there, serotonin is released into other parts of the brain to influence different behaviors. Although drugs that target serotonin are widely used as antidepressants, how this chemical signal acts in the brain remains a mystery. This is due, in part, to it being technically challenging to carry out experiments on the serotonin-producing neurons.

A technique called optogenetics uses light to selectively activate or inhibit individual cells in live animals. Here, Correia, Lottem et al. use optogenetics to activate serotonin-producing neurons in the dorsal raphe nucleus of mice. The experiments show that triggering serotonin production for a few seconds causes the mice to move around more slowly as they explore their surroundings. This short-term release of serotonin only slows the mice down if they are not already occupied with other activities, such as finding water or balancing on a moving object. These experiments suggest that serotonin decreases an individual's motivation to move but that this can be overcome by sufficiently powerful goals. In contrast, repeatedly activating the serotonin neurons over a period of several weeks led to long-term changes of the opposite kind – the mice begin to move around more quickly.

The findings of Correia, Lottem et al. have possible implications for the use of drugs that target serotonin to treat mental disorders as it suggests important links between serotonin, movement, and the ability of the brain to change how it responds to certain situations. The next steps will be to investigate how the two different effects of serotonin are connected, which areas in the brain are involved and how best to apply these findings to clinical studies.

tonic 5-HT have been proposed as well (*Daw et al., 2002*), but little is in fact known about the effects of phasic 5-HT firing. Optogenetic manipulations, which afford both genetic specificity and temporal control, might access phasic effects that are inaccessible by pharmacological manipulation (e.g., *Tsai et al., 2009*).

Here, we used an optogenetic approach to target 5-HT neurons in the dorsal raphe nucleus (DRN), the main source of 5-HT to the forebrain, for activation. Our starting point was experiments in the open field arena, a widely used assay for spontaneous behavior (*Hall, 1934*; *Seibenhener and Wooten, 2015*). A long-standing theory of 5-HT suggests that it is a mediator of 'behavioral inhibition' (*Soubrié, 1986*). This theory was motivated by data showing that 5-HT depletion increases startle responses (*Davis and Sheard, 1974*; *Davis et al., 1980*) and locomotor activity (*Gately et al., 1985*; *Eagle et al., 2009*). On the other hand, it has also been theorized, largely based on recordings of 5-HT neurons, that the primary function of the 5-HT system is instead the facilitation of rhythmic motor behaviors (*Jacobs and Fornal, 1993*). We found that phasic activation of DRN 5-HT neurons caused a rapid and striking decrease in locomotion, an effect of order 50%, while sparing other behaviors such as grooming.

We then examined the detailed kinematics of locomotor gait using a linear track (*Machado et al., 2015*). Surprisingly, in this assay we saw no effect of 5-HT activation, arguing against a motor hypothesis and suggesting a higher level underlying cause leading to motor effects. Indeed, 5-HT is strongly implicated in the control of impulsivity (*Dalley and Roiser, 2012*), suppressing actions because it helps avoiding future punishments (*Dayan and Huys, 2008*, *2009*) or obtaining delayed rewards (*Miyazaki et al., 2011a*, *2011b*, *2012*, *2014*; *Fonseca et al., 2015*). In either case, serotonin is thought to drive behavioral inhibition by altering the impact of the future motivating outcome. Anxiety is a prominent motivation in the open field, and mice show strong center-avoidance or thigmotaxis (reviewed in *Prut and Belzung (2003)*). Yet, we found effects on locomotion that were not accompanied by spatial biases and yielded no positive or negative effects on anxiety measures.

It has also been theorized that 5-HT signals worse-than expected outcomes and might drive aversive learning (*Daw et al., 2002*; *Dayan and Huys, 2009*). Conversely, 5-HT has also been proposed

to signal reward (*Liu et al., 2014*). To test if phasic 5-HT encodes or modulates reward or punishment, we repeatedly paired phasic 5-HT stimulation with a specific location within the open field, but we found neither appetitive or aversive place preference learning. Instead we found that slowing of movement could masquerade as a place preference depending on how occupancy was assessed. These findings support previous studies showing that DRN 5-HT activation does not itself cause reinforcement learning (*Miyazaki et al., 2014*; *Fonseca et al., 2015*). Unexpectedly, however, we found that repeated daily phasic 5-HT activation resulted in a long-term enhancement of locomotion, an effect similar in magnitude to the transient effect but opposite in sign. This observation recalls the two-to-three week window for selective serotonin re-uptake inhibitors (SSRI) therapeutic effects to develop (*Machado-Vieira et al., 2010*) and studies linking 5-HT activation to plasticity (*Santarelli et al., 2003*; *Maya Vetencourt et al., 2008*).

## Results

### Optogenetic activation of DRN 5-HT neurons reduces spontaneous locomotion in the open field

To activate DRN 5-HT neurons, we expressed the light-sensitive ion channel channelrhodopsin-2 (ChR2) in DRN 5-HT neurons using an AAV2/9 viral vector (AAV2/9-Dio-ChR2-EYFP) injected into the DRN of SERT-Cre mice or wild-type littermate controls (WT) and implanted an optical fiber in the same location (*Figure 1A*) (see *Dugué et al. (2014)* for more details). Experimenters were blind to the mice's genotype throughout training and testing in this and all experiments below. Histological analysis performed at the end of testing confirmed ChR2-YFP expression was localized to the DRN in SERT-Cre animals (*Figure 1B*) and that there was no expression in WT controls (data not shown).

We first tested mice in an open field assay in which the effect of stimulation on relatively spontaneous behaviors (not driven by overt reward or punishment) could be characterized (*Figure 1C*). SERT-Cre and WT mice received blue light pulses in alternating 5 min blocks with 3 s light on, interleaved with 7 s light off, during stimulation blocks (*Figure 1D*). To characterize the effects of DRN 5-HT activation we first performed a video based ethological analysis (*Choleris et al., 2001*; *Video 1*). The most frequent behaviors observed in the open field were walking and rearing, with the remainder of time spent grooming, digging or otherwise awake but relatively stationary (*Figure 1E*). Scratching and jumping were observed very infrequently and were not further analyzed. DRN 5-HT activation induced a profound decrease in locomotion, (p=3.61 $\times$ 10$^{-5}$, paired t-test, SERT-Cre mice, N = 15), *Figure 1E*) and a robust increase in the probability of the 'resting' state (p=4.75 $\times$ 10$^{-3}$, paired t-test, SERT-Cre mice, N = 15), *Figure 1E*). Rearing was also reduced (p=0.0495), while digging and grooming were unaffected (*Figure 1E*, digging, p=0.536; grooming, p=0.582; paired t-test, SERT-Cre mice, N = 15). The 'resting' state did not correspond to freezing, as the animals did not exhibit a crouching position and often continued to make small movements (*Blanchard and Blanchard, 1969*), sometimes accompanied by a lowering of the head (*Video 1*). To summarize the dynamics of DRN 5-HT activation effects on behavioral state, we grouped the 'mobile' states (walking, jumping, rearing) and the relatively 'immobile' (resting, grooming, digging, scratching) and plotted their probability as a function of time relative to stimulation (*Figure 1F–G*). This showed that the onset of DRN 5-HT activation effects were very rapid, peaking within 2 s and returning to baseline within 5 s (*Figure 1F–G*).

### Optogenetic activation of DRN 5-HT neurons rapidly suppresses locomotion speed

To better quantify these effects in a simple and robust manner, we extracted the mouse's spatial position over time from video data using standard automated tracking methods, allowing us to obtain more precise measurements over a larger data set (see Materials and methods). Optogenetic activation occurred heterogeneously in space, as illustrated by the position traces for representative WT and SERT-Cre mice (*Figure 2A*). As suggested by the ethological analysis, light delivery in SERT-Cre, but not WT, mice resulted in a rapid decrease in movement speed, with >90% of the decrease being reached within <1 s (*Figure 2B*). The reduction in average speed was clearly visible for every SERT-Cre mouse tested and highly robust at the group level (6.14 ± 0.31 cm·s$^{-1}$ pre-stimulation and 2.94 ± 0.24 cm·s$^{-1}$ post-stimulation (mean ± SEM); p=4.80 $\times$ 10$^{-7}$, paired t-test, SERT-Cre mice,

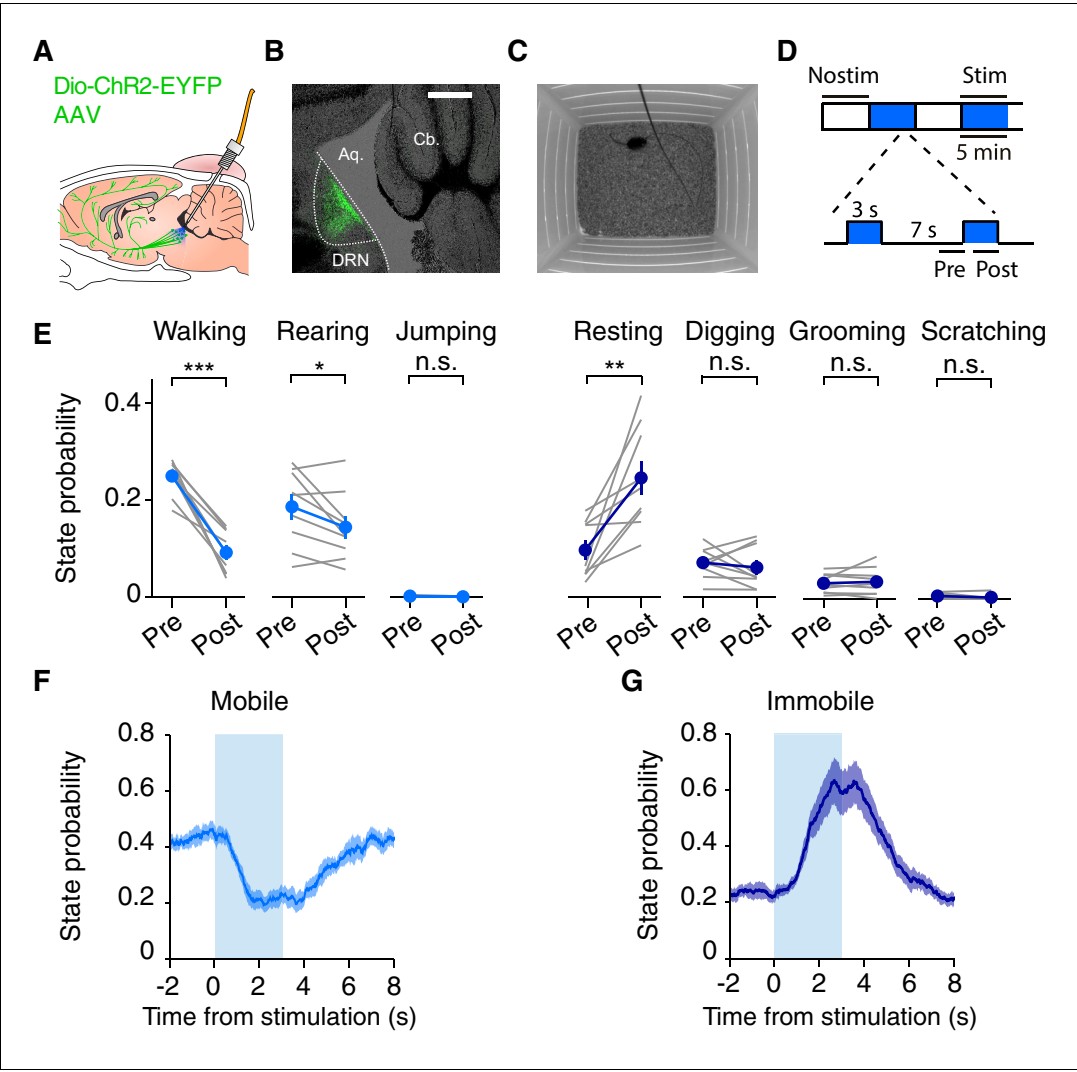

**Figure 1.** Optogenetic DRN 5-HT activation reduces spontaneous locomotion in the open field. (A) Schematic of the optogenetic approach. DRN neurons are infected with AAV2/9-Dio-ChR2-EYFP. In SERT-Cre mice, 5-HT neurons will express ChR2-YFP (green cells) and can be photoactivated with blue light delivered through an implanted optical fiber. (B) Fluorescence image of a parasagittal section showing ChR2-YFP expression (green) localized to the DRN. Scale bar, 500 μm. (C) Schematic drawing of the open field paradigm. (D) Schematic diagram of photostimulation protocol. Each 30 min session consisted of stimulated (stim, blue) and non-stimulated (nostim, white) blocks of 5 min. A session always starting with a non-stimulated block and blocks always alternated, for a total of 30 min. During stim blocks, 3 s pulse trains of light were delivered every 10 s. Pre and post intervals shown were used to calculate stimulation effects. (E) Probability of being in a specific behavioral state for non-stimulated (pre) and stimulated (post) intervals for the population of SERT-Cre mice (N = 15). Note that probabilities do not sum to 100% because scoring does not include all time points. Individual mice shown in grey lines and averages across mice in filled circles. Error bars indicate SEM. In some cases, the error bars are too small to be visible. n.s: not significant. *p<0.05, **p<0.01, ***p<0.001, with paired t-test. (F) Probability of being in a mobile state (walking, rearing, jumping), as a function of time relative to stimulation onset. The shaded area indicates SEM across mice. (G) Same as (F) but for the immobile states (resting, digging, grooming and scratching).

N = 15 mice, *Figure 2C*). The effect of stimulation could be seen as a shift in the distribution of movement speeds toward slower speeds (*Figure 2D*). The difference between post- and pre-stimulation speed was also highly significantly different between SERT-Cre and WT animals (t-test, p=7.45 × 10⁻⁷, *Figure 2E*). The effect of optogenetic activation was dose-dependent over a range of lower

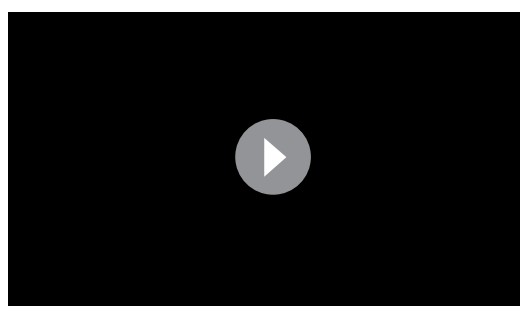

**Video 1.** Example SERT-Cre mouse behavior in the open field experiment during a photostimulation block.   Also shown are the behavioral state (red), photostimulation condition (on/off, filled/empty blue square respectively), and speed (green rectangle, normalized to the maximum value in that session).

stimulation frequencies (5, 15 and 25 Hz; Pearson correlation coefficient, r = −0.837, p=6.84 × 10$^{-4}$, *Figure 2F*). No significant effects were found in any measure for WT mice (N = 9).

## Effect of DRN 5-HT activation is largely independent of previous movement speed

These results indicate that optogenetic activation of DRN 5-HT neurons rapidly switches animals away from engaging in active behaviors such as locomotion or rearing and suggest that they may slow the speed of locomotion. We next examined whether these effects were specific to the state of the animal at the onset of stimulation. We first tested whether the effects depend on the prior speed of the animal by conditioning the speed plots by the pre-stimulation speed (*Figure 2G*). This conditioning procedure produces a tendency of the speed to revert back toward the mean, which can be seen even in non-stimulated blocks (*Figure 2H–I*). We therefore calculated the stimulation effect by subtracting speed values in equivalent non-stimulation blocks from values in the stimulation blocks ('delta speed', *Figure 2J*). There were significant effects on the average delta speed (post-pre intervals) for all pre-stimulation speeds (*Figure 2K*), as indicated by comparing delta speed between non-stimulated and stimulated trials across different speed quartiles (2-way ANOVA, stimulation, prior speed; SERT-Cre mice, N = 15; stimulation, $F_{(1,110)}$=123, p=1.18 × 10$^{-4}$; prior speed, $F_{(3,110)}$=160, p=6.53 × 10$^{-40}$; stimulation x prior speed $F_{(3,110)}$=18.3, p=1.02 × 10$^{-9}$), followed by paired t-tests, with Bonferroni correction (stimulation, p<0.05; prior speed quartiles 1–2, n.s; other quartiles, p<0.05). This analysis shows that decrease in speed produced by DRN 5-HT neuron activation was larger when the speed of the animal just prior to stimulation onset was faster. No significant changes were found for WT mice (data not shown, N = 9). This indicates that the effects of optogenetic activation of DRN 5-HT neurons were not limited to biasing animals against initiating movement, but also rapidly stopped mice even when they were already moving at relatively high speed.

## DRN 5-HT activation does not induce general motor impairment

These observations suggest an unconditional effect on active motor behavior and indicate a possible impairment in motor coordination during active movements. To test for possible effects of optogenetic activation of DRN 5-HT neurons on motor output and coordination, we first used the accelerating rotarod assay (*Figure 3A*), a widely used assay for motor coordination in rodents (*Carter et al., 2001*). After allowing 2–3 weeks for virus expression, mice were trained for two consecutive days on the rotarod and on the third day DRN 5-HT neurons were optogentically activated (5 mW, 10 ms, 490 nm light pulses at 25 Hz) in a randomly-selected 50% of trials, i.e. from placement of the mouse on the rotarod until the mouse fell (*Figure 3B–C*). We found no difference in latency to fall between stimulated and non-stimulated trials for SERT-Cre mice (p=0.743, paired t-test, N = 7) or between WT and SERT-Cre mice (stim. and non-stim. trials, p=0.513, two-sample t-test, SERT-Cre mice, N = 7, WT mice, N = 5, *Figure 3D*). Thus optogenetic stimulation was without obvious effects on motor coordination assessed by this test.

It is possible that the rotarod assay failed to reveal a more subtle effect on motor coordination. Therefore we next studied a subset of mice using a linear track assay with high-speed videography, allowing tracking the trajectory of the four paws, nose and tail (LocoMouse system, *Machado et al., 2015*). In this assay, mice cross back and forth in a narrow brightly-lit corridor (*Figure 4A–B*). Mice are motivated to cross the track by the tendency to avoid light and by water rewards they received at either side (mice were mildly water-restricted). DRN 5-HT optogenetic activation was delivered starting when the animal entered the track and until it reached the other end (stimulation parameters as in the rotarod assay). Surprisingly, despite the effects on locomotor behavior in the open field, we

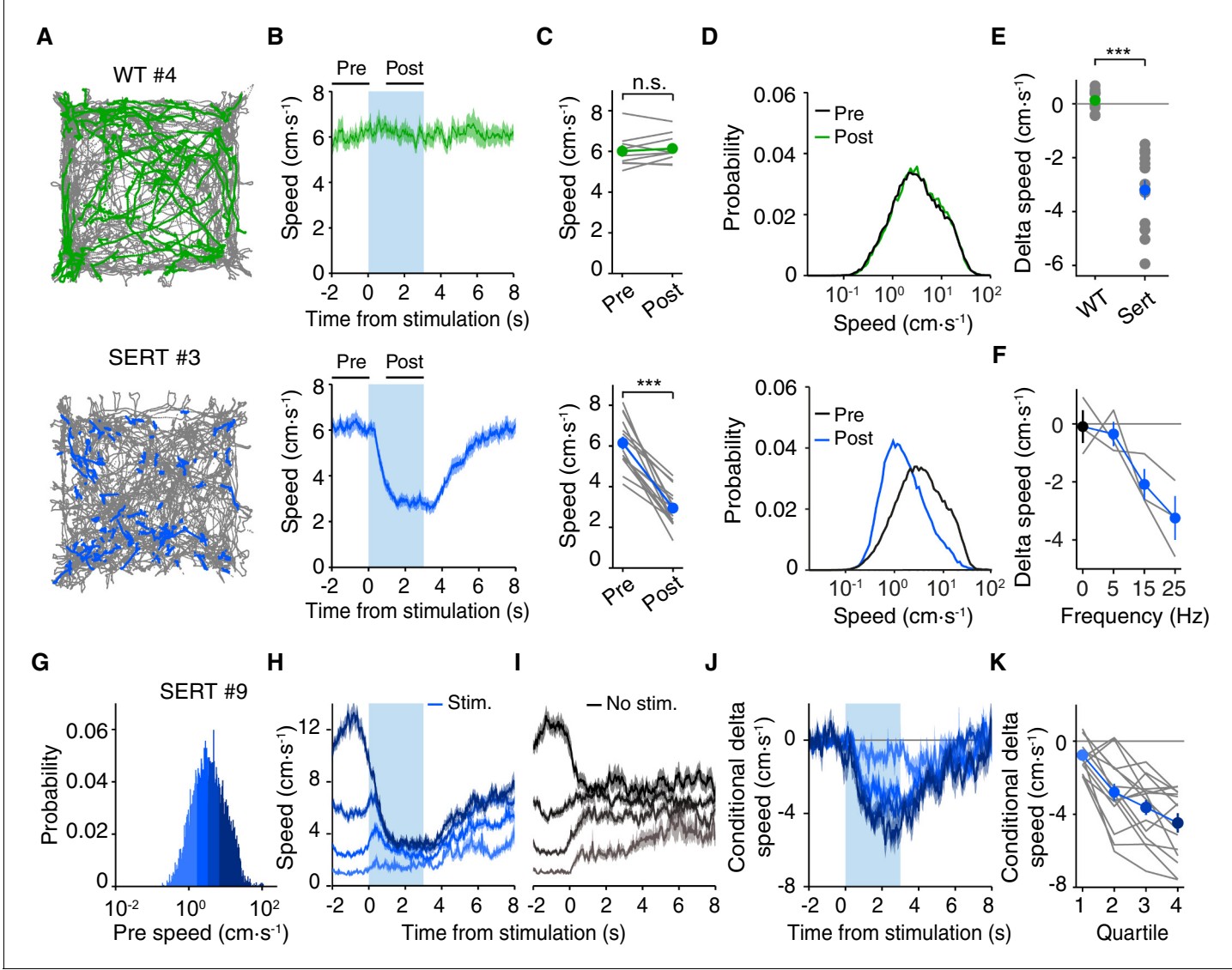

**Figure 2.** Optogenetic DRN 5-HT activation slows down animals in the open field, independently of previous locomotion speed. (A) Position tracking of example WT (top) and SERT-Cre (bottom) mice. All positions visited in the session are shown in gray; the positions visited during each 3 s stimulation periods are shown for one WT (green) and one SERT-Cre (blue) mouse. (B) Time course or speed relative to stimulation onset. Here, and below, WT (N = 9) are shown in green and SERT-Cre mice (N = 15) in blue and data mean ± SEM across animals is shown. Pre and post intervals are indicated by horizontal lines. (C) Average speed in pre- and post-stimulation intervals for individual mice (gray lines) and for the population of mice (mean ± SEM). In some cases, the error bars are too small to be visible. n.s: not significant. ***p<0.001 with paired t test. (D) Probability distribution of speed in pre- and post-stimulation intervals for the population of WT and SERT-Cre mice. (E) Difference between speed in post- and pre-stimulation intervals (delta speed) ***, p<0.001 with two-sample t test. (F) Dependence of delta speed on frequency of stimulation for the individual mice (gray lines) and for the subset of SERT-Cre mice tested (mean ± SEM, N = 3). (G) Speed probability distribution in the pre interval for all stimulation periods within stimulated blocks (as well as equivalent measures for non-stimulated blocks) for an example SERT-Cre mouse. (H) Average post speed in stimulated blocks, conditioned by pre speed for SERT-Cre mice. The four colors indicate the speed ranges used for pre conditioning, as indicated in the distribution in (G). (I) The same as (H) but for equivalent period in non-stimulated blocks. (J) The difference between the stimulated and non-stimulated blocks shows the effect of stimulation conditioned on pre speed. (K) Average difference between stimulated and non-stimulated blocks for delta speed (difference between post- and pre-intervals) for each quartile for individual mice (gray lines) and for the population of SERT-Cre mice (mean ± SEM, N = 15). In some cases, the error bars are too small to be visible.

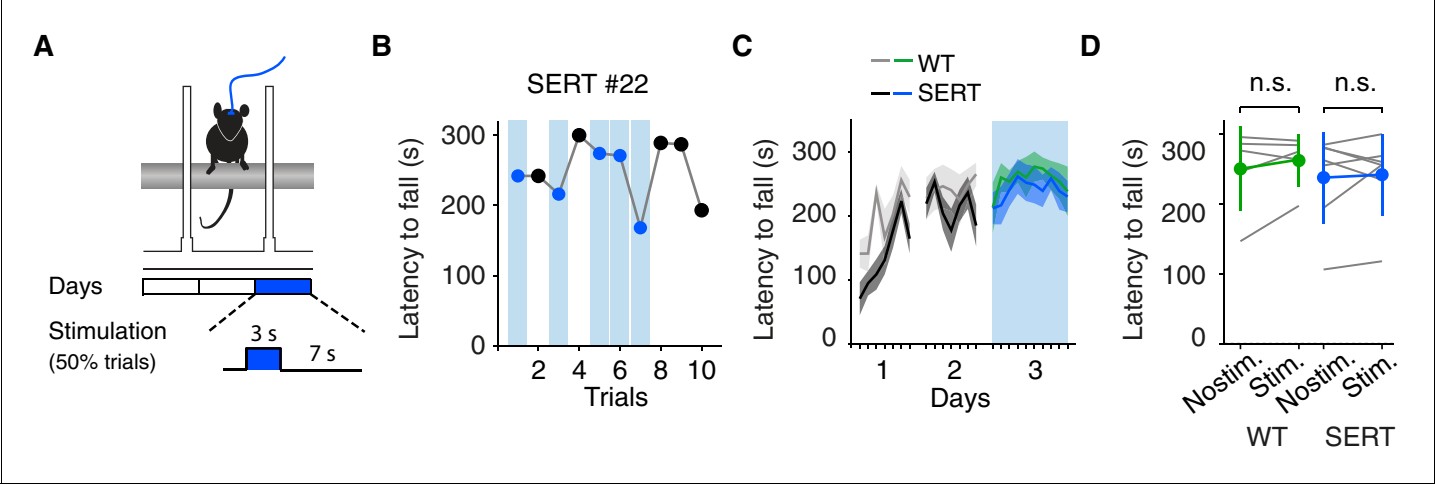

**Figure 3.** Optogenetic DRN 5-HT activation does not affect motor coordination in the rotarod assay. (**A**) Schematic of the accelerating rotarod assay. (**B**) Latency to fall for one example SERT-Cre mouse, with randomly interspersed stimulated trials. (**C**) Average latency to fall, showing learning and testing period. Note, stimulation (cyan bar) only occurred after training, WT (green, n = 5) and SERT-Cre (blue, N = 7) mice (mean ± SEM) are shown. (**D**) Average latency to fall in the testing session for individual mice (gray lines) and for the population of WT mice (green, N = 5) and SERT-Cre mice (blue, N = 7, mean ± SEM). In some cases, the error bars are too small to be visible. n.s: not significant.

observed no effect of stimulation on speed, measured either as time to cross the track (p=0.446, paired t-test, SERT-Cre mice, N = 7, *Figure 4C*) or the speed (p=0.387, paired t-test, SERT-Cre mice, N = 7, *Figure 4D*), for the population of SERT-Cre mice. We also ran the same mice in our standard open field protocol (*Figures 1–2*) and found a similar magnitude of effects on speed as described above (pre-speed 7.32 ± 0.73 cm·s$^{-1}$, post-speed 5.39 ± 0.65, p=9.63 × 10$^{-4}$, paired t-test, SERT-Cre mice, N = 7).

To test for more subtle possible effects, we next analyzed detailed locomotor kinematics (*Figure 4A–B*). We performed single-limb gait analyses, characterizing continuous 3D paw trajectories in forward (x-axis), side-to-side (y-axis) and vertical (z-axis) directions, for different speed intervals. We found no difference in peak swing velocity between stimulated and non-stimulated trials (p=0.878, paired t-test, SERT-Cre mice, N = 7, *Figure 4E*). Neither the base of support between stimulated and non-stimulated trials (p=0.586, paired t-test, SERT-Cre mice, N = 7, *Figure 4F*), nor the vertical (z) trajectory peak (p=0.723, paired t-test, SERT-Cre mice, N = 7, *Figure 4G*) were affected by stimulation. Polar plots indicating the phase of the step cycle in which each limb enters stance (aligned to stance onset of the front right paw), showed no differences between DRN 5-HT activated and non-activated trials (3-way ANOVA with paw, speed and stimulation as main factors; stimulation: F(185,1)=0.209, p=0.648, SERT-Cre mice, N = 7, *Figure 4—figure supplement 1A*). Furthermore, neither vertical (z) movement of the nose (peak values) (p=0.641, paired t-test, SERT-Cre mice, N = 7, *Figure 4—figure supplement 1B*) nor tail side-to-side oscillations (y) (p=0.310, paired t-test, SERT-Cre mice, N = 7, *Figure 4—figure supplement 1C*) trajectories were affected in stimulated trials.

## DRN 5-HT photostimulation does not induce anxiety-like behavior

The results with the linear track suggest that DRN 5-HT optogenetic activation does not effect motor coordination or the detailed kinematics of locomotor behavior. However, this is secondary to the unanticipated finding that the primary effect on locomotor speed is highly context-dependent. This might be attributed to the strong drive to perform specific actions in those contexts. Therefore, we next turned to consider whether the effects on locomotion could be a consequence of specific affective or motivational states that might be induced by optogenetic activation of DRN 5-HT neurons. The open field assay has long been used as a screen for anxiety-related behavior in rodents (*Bailey and Crawley, 2009*). Thigmotaxis, the tendency to remain close to the walls and avoid the center of the arena, is a widely used measure of anxiety with ethological face validity

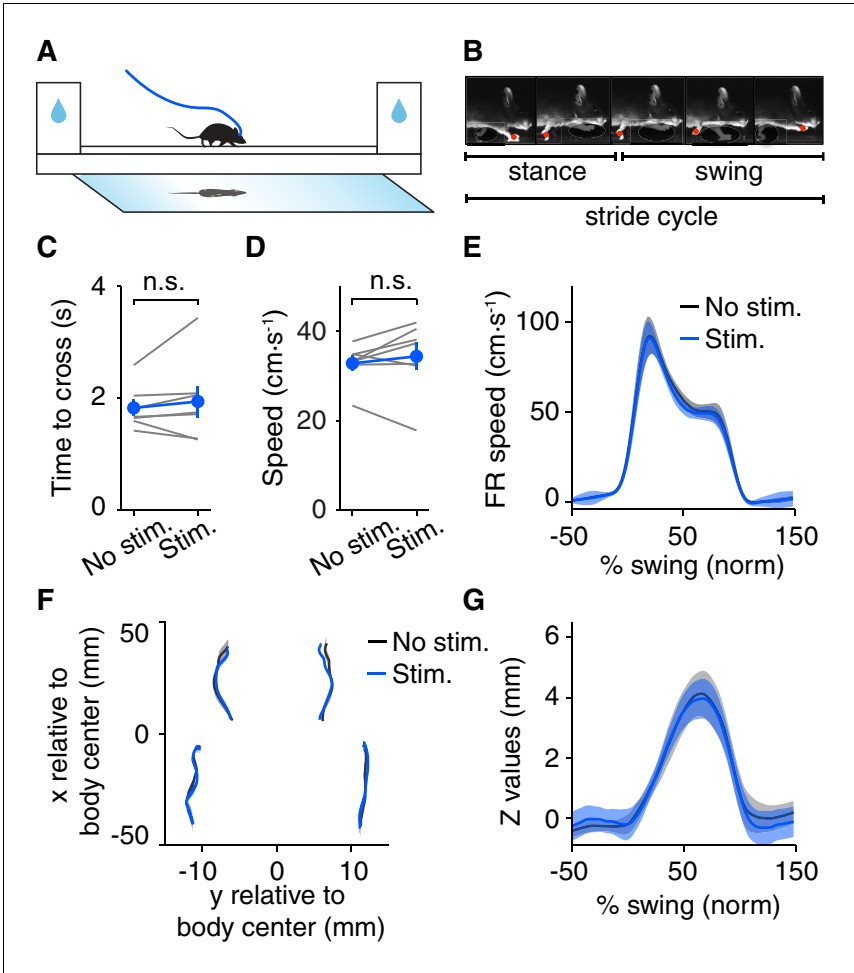

**Figure 4.** Optogenetic DRN 5-HT activation does not induce motor impairment in the LocoMouse assay. (**A**) Schematic drawing of the LocoMouse apparatus. Water deprived animals walk freely across a glass corridor connected to two boxes with water ports. A mirror below at 45° angle allows a single high-speed camera to capture side and bottom views at 400 frames per second. DRN 5-HT photostimulation occured randomly in 50% of crossings. (**B**) Paws, nose and tail segments were automatically segmented and tracked in 3D. Individual strides were divided into swing and stance phases for further analysis. (**C**) Average time to cross the linear track for individual mice (gray lines) and for the population of SERT-Cre mice (N = 7, mean ± SEM) in non-stimulated (black) and stimulated (blue) trials. In some cases, the error bars are too small to be visible. n.s: not significant. (**D**) Average whole-body speed (center of mass) for individual mice (gray lines) and for the population of SERT-Cre mice (N = 7, mean ± SEM) in non-stimulated (black) and stimulated (blue) trials. In some cases, the error bars are too small to be visible. n.s: not significant. (**E**) Instantaneous forward speed of front-right paw during swing phase at stride speed of 15–20 cm·s$^{-1}$ for stimulated (blue) and non-stimulated (black) crossings. (**F**) x-y position of four paws relative to the body center during swing. (**G**) Vertical (z) position of front-right paw relative to ground during swing.

The following figure supplement is available for figure 4:

**Figure supplement 1.** DRN 5-HT activation does not affect motor coordination and locomotion.

---

(*Choleris et al., 2001*). If the observed locomotor effects were secondary to an increase (or decrease) in fear or anxiety they should be accompanied by changes in thigmotaxis or by more subtle interaction between location and stimulation effects.

We first characterized the effect of DRN 5-HT activation conditioned on the location of the mouse within the open field arena at the time of stimulation onset (center, periphery and corners; *Figure 5A*). Mice spent more time in the corners and periphery and less in the center (*Prut and*

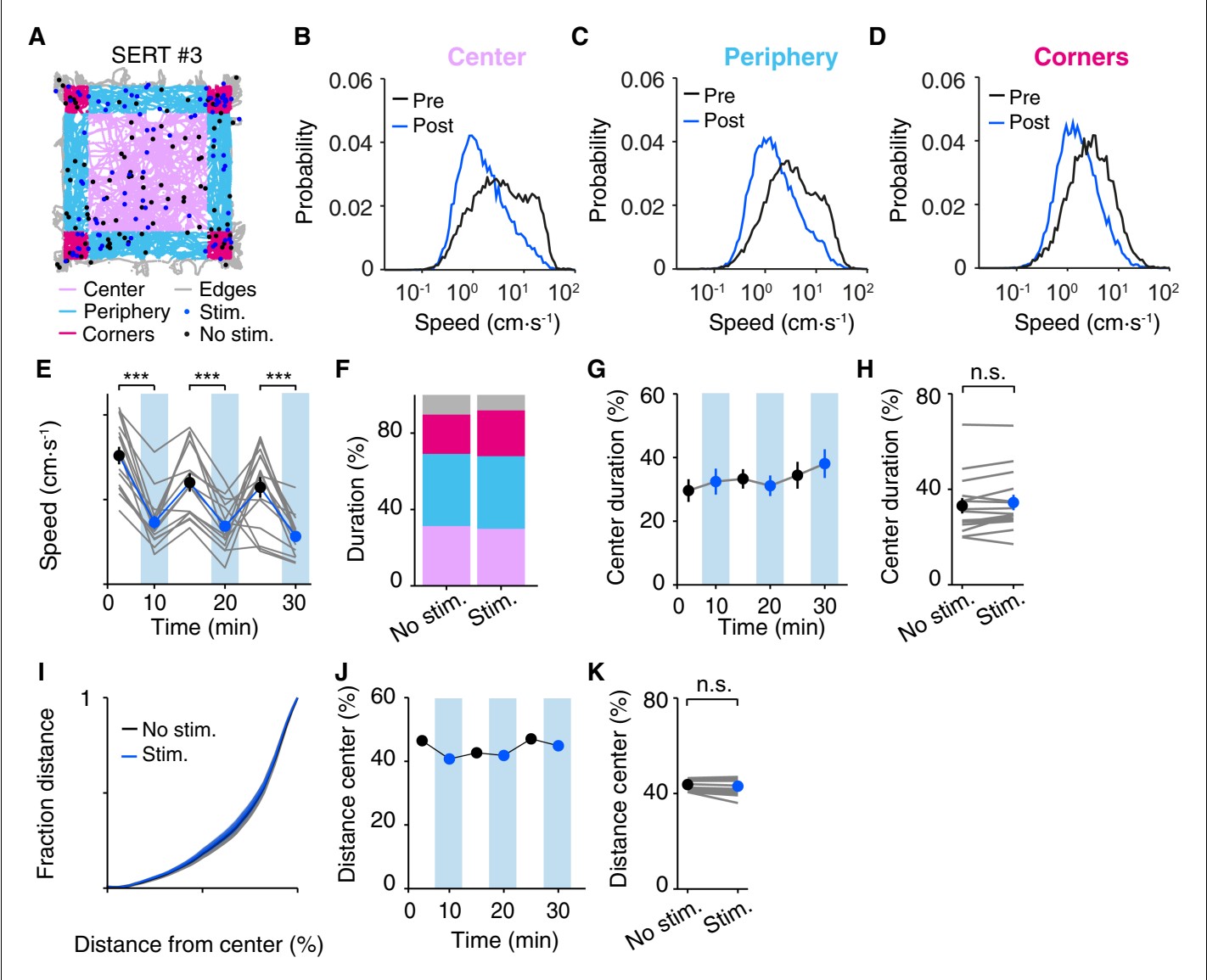

**Figure 5.** Effect of DRN 5-HT optogenetic activation does not induce anxiety-like behavior in the open field. (A) Position tracks of an example SERT-Cre mouse, depicting the main areas of the open field: corners (dark pink), periphery (light blue), center (light pink) and edges (gray). Filled circles represent the position of the mouse at the beginning of each stimulus train. For this and subsequent panels, blue indicates stimulated blocks and black indicates equivalent times in non-stimulated blocks. (B) Speed probability distributions during pre and post stimulation intervals that began when the mouse was in the center area of the open field mice. For this and subsequent panels, data is pooled or averaged across all SERT-Cre mice (N = 15). (C, D) The same as (B) but for the periphery and corners areas. (E) Average speed across blocks within a session for individual mice (gray lines) and for the population of SERT-Cre mice (N = 15, mean ± SEM) in non-stimulated (black) and stimulated (blue) blocks. In some cases, the error bars are too small to be visible. Each pair of stimulated and non-stimulated block was compared (***p<10$^{-3}$, paired t test with Bonferroni correction for multiple comparisons). (F) Fractional occupancy in each area of the open field, for the population of SERT-Cre mice (N = 15), color code as in (A). (G) Fraction center area occupancy as a function of duration within the session. (H) Same as G but showing individual mice averaged over the entire session duration. (I) The cumulative distribution of the average distance from the geometric center point of the open field normalized to the distance of the walls to the center. (J) Fraction of the total distance travelled that was in the center area as function of time within the session. (K) Same as J but showing individual mice averaged over the entire session duration.

*Belzung, 2003*) and showed distinct speed probability distributions depending on their location within the arena, with higher speeds mainly being exhibited in the center (*Figure 5B–D*). Nevertheless, DRN 5-HT activation caused a robust decrease in speed regardless of the spatial area where the animal was located at stimulation onset (*Figure 5B–D*), as indicated by comparing speed

between pre- and post-stimulation intervals across the different areas (2-way ANOVA, stimulation, areas; SERT-Cre mice, N = 15; stimulation, $F_{(1,84)}$=250, p=2.58 × $10^{-17}$; areas, $F_{(2,84)}$=18.9, p=3.99 × $10^{-4}$; stimulation x areas, $F_{(2,84)}$=8.58, p=0.0238). The effect was confirmed with paired t-tests comparing speed pre- and post-stimulation (center, p=6.19 × $10^{-5}$; periphery, p=2.50 × $10^{-7}$; corners p=6.08 × $10^{-7}$; SERT-Cre mice, N = 15). Given the different behaviors and speeds in each area it is not surprising that there was a positive interaction between stimulation and areas factors. We found no significant effects between stimulated and non-stimulated trials across areas for WT animals (data not shown, N = 9).

Measures of anxiety may be sensitive to the amount of exposure to the environment, as animals become more accustomed to it. We next examined whether stimulation effects were dependent on the amount of time the animal was in the arena (*Figure 5F*). Animals showed a tendency to move more slowly with time spent in the box over 30 min (*Figure 5E*). Optogenetic activation caused a decrease in speed in SERT-Cre mice across all 5 min blocks (*Figure 5E*) as indicated by a 2-way ANOVA (stimulation, blocks; SERT-Cre mice, N = 15; stimulation, $F_{(1,84)}$=67.6, p=0.0218; blocks, $F_{(2,84)}$=4.36, p=0.0158; stimulation x blocks, $F_{(2,84)}$=1.18, p=0.314), followed by paired t-tests for comparison between non-stimulated and stimulated trials across blocks (blocks 1–3, p<0.001). There were no significant effects for WT animals (data not shown, N = 9).

The above results show that optogenetic activation of DRN 5-HT neurons is insensitive to the state of the animal when it occurs within the open field, but is somewhat more robust in its effects on rapid excursions (*Figure 2H–K*) that occur primarily in the center of the box (*Figure 5B*). Taken together, these observations might suggest that DRN 5-HT activation could tend to decrease the frequency or duration of center excursions, increasing thigmotaxis. However, we observed no change in center occupancy between stimulated and non-stimulated blocks (2-way ANOVAs, with block and stimulation condition; stimulation, $F_{(1,2)}$=0.611, p=0.516; blocks, $F_{(2,2)}$=3.08, p=0.245, SERT-Cre, N = 15; *Figure 5G*). A post-hoc comparison of total center occupancy between stimulated and non-stimulated blocks again showed no difference (p=0.763, paired t-test, *Figure 5H*). Similarly, we observed no effect in two other measures of thigmotaxis: distance from the center of the open field arena (p>0.999, Kolmogorov-Smirnov test comparing stim. and non-stim. distributions, *Figure 5I*) and ratio of central vs. peripheral movement distance (2-way ANOVA, with block and stimulation condition; stimulation, $F_{(1,2)}$=4.01, p=0.183; blocks, $F_{(2,2)}$=2.22, p=0.311, SERT-Cre, N = 15, *Figure 5J* and post-hoc comparison of the ratio of central vs. peripheral distance between stimulated and non-stimulated blocks, p=0.139, paired t-test, *Figure 5K*). No significant changes were observed in any measure for WT mice (data not shown, N = 9). Thus, photostimulation effects were both largely independent of the mouse's location and while they biased action selection and speed, they did not bias mice toward or away from the center or periphery of the arena.

## Spatially-specific DRN 5-HT optogenetic activation does not yield place preference or avoidance

These results show that DRN 5-HT activation does not affect thigmotaxis in the open field arena on the time scale of seconds to minutes. Therefore the effects of phasic optogenetic activation of these neurons do not appear to result from attenuation or enhancement of anxiety insofar as that can be measured by center avoidance. However, they do not exclude the possibility of longer-term effects. For example, some important affective and motivational effects of the dopamine system build up over longer periods and require contingency with environmental events, such as the conditioned place preference that can be produced by stimulating dopamine neurons contingent on an animal being in a certain location (*McDevitt et al., 2014*; *Tsai et al., 2009*). Indeed, at least one report has reported reinforcing effects of DRN 5-HT activation based on a 'real time' conditioned optogenetic stimulation assay (*Liu et al., 2014*).

In order to test for such effects, we ran an open field assay with photostimulation delivered contingent on the mouse entering a spatial region of interest (ROI) within the arena (*Figure 6*). Following *Lui et al. (2014)*, we defined the ROI as a sub-region of the center area (*Figure 6A*); animals had no visible cues for the ROI. On each day mice freely explored the arena for 15 min. After one habituation session, SERT-Cre (N = 4) and WT (N = 5) mice were subjected to three days of conditioning, with photostimulation being delivered during the entire excursion within the defined ROI (10 ms, 20 mW pulses at 25 Hz). They were then tested for one day without photostimulation. Note that we used this increased laser power and frequency of pulses in order to match the protocol of *Liu et al.*

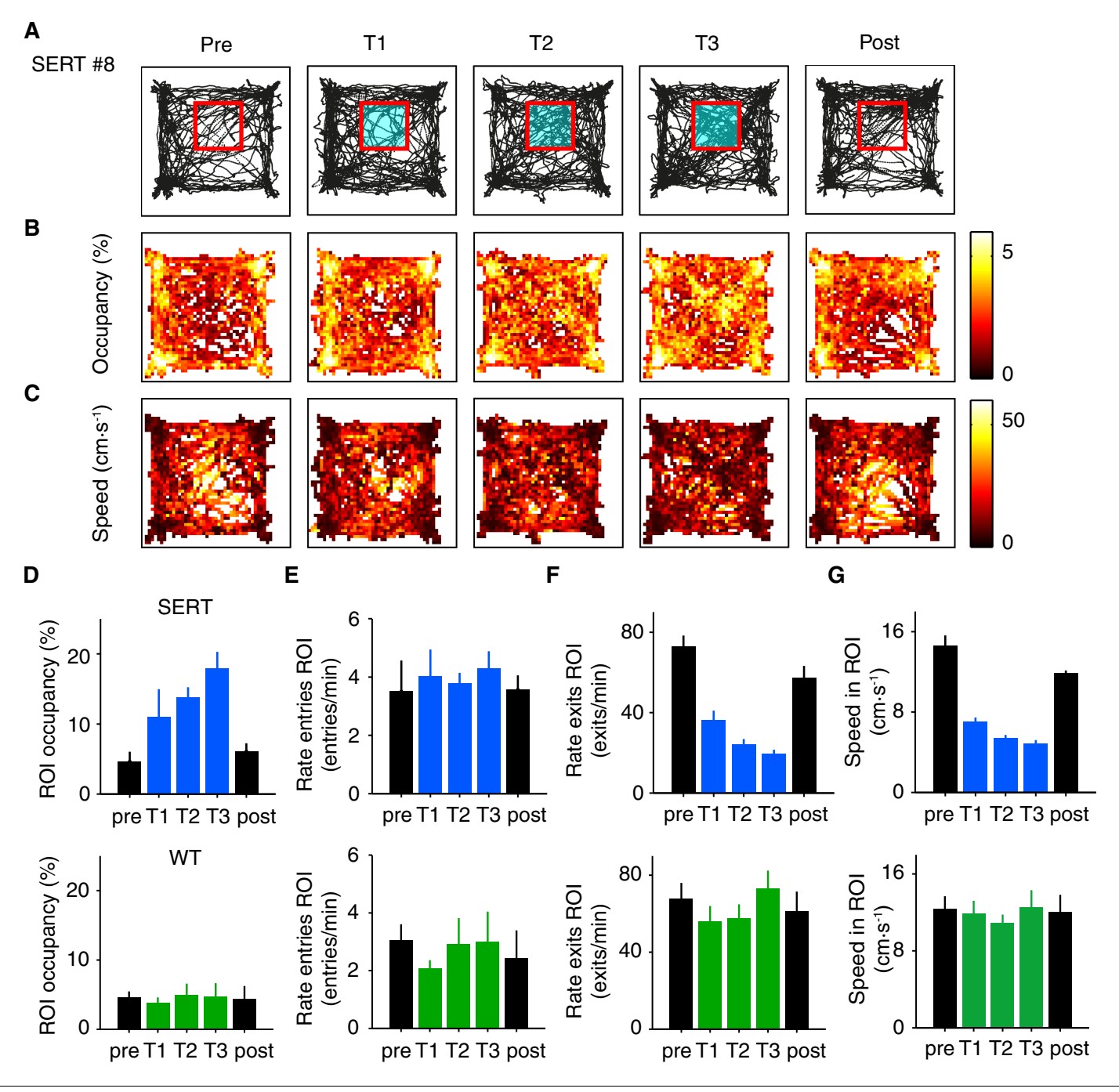

**Figure 6.** Optogenetic DRN 5-HT activation in a specific region of interest does not produce aversive or appetitive responses. (**A**) Position tracks for an example SERT-Cre mouse for sessions before (pre), during (T1, T2, T3) and after (post) photostimulation. Red square indicates the ROI in which photostimulation occurred (indicted by blue lines). (**B**, **C**) Heat maps depicting the normalized occupancy and average speed for the same sessions depicted in **A**. (**D**–**G**) Population data (mean ± SEM) for SERT-Cre mice (N = 4; above, blue are stimulated sessions) or WT mice (N = 5; below, green are stimulated sessions). Different measures as indicated.

*(2014)*. We also ran a subset of mice in our standard open field protocol (*Figures 1–2*), in which stimulation occurs in random locations, using the same amplitude and found a similar magnitude of effects on speed (pre-speed 7.4 ± 0.67 cm·s$^{-1}$, post-speed 4.0 ± 0.16 cm·s$^{-1}$, p=0.0103, paired t-test,

SERT-Cre mice, N = 4). Therefore we believe the ROI and standard protocol are comparable except for the presence of spatial contingency of stimulation in the former.

Given the decrease of speed induced by optogenetic activation of DRN 5-HT neurons in the standard protocol, we expected that animals would spend more time in the ROI simply as a consequence of slowing. Locomotion traces, occupancy and speed heat maps for all five days of the assay for a representative SERT-Cre mouse are shown in *Figure 6A–C*. Optogenetic activation of DRN 5-HT neurons produced a significant increase in ROI occupancy ($F_{(4,15)}$=5.56, p=6.00 $\times$ 10$^{-3}$, 1-way ANOVA across days; *Figure 6D*). Consistent with a movement slowing effect, we found a decrease in the average speed inside the ROI ($F_{(4,15)}$=68.9, p=1.80 $\times$ 10$^{-9}$, ANOVA; *Figure 6G*) and the rate of exits from the ROI ($F_{(4,15)}$=26.6, p=1.17 $\times$ 10$^{-6}$, ANOVA; *Figure 6F*) across stimulated sessions for the population of SERT-Cre mice (N = 4). Post-hoc tests comparing the effect of each significant variable across sessions for SERT-Cre animals showed a significant increase in ROI occupancy by the third stimulated session and increases in rate of exits from the ROI and average speed in the ROI by the first stimulated session (*Figure 6D,F,G*, p<0.05, N = 4, t-tests with Bonferroni correction). No significant effects were observed in WT mice (N = 5, *Figure 6D–G* bottom) and no significant differences were found for average speed outside the ROI across sessions for either SERT-Cre or WT mice (data not shown).

These effects might be interpreted as either a place preference or as a stimulation-dependent reduction in movement speed. However, two additional observations argue for the latter interpretation. First, if stimulation reinforced the value of the ROI, we would expect mice to return more frequently to it. Contrary to this expectation, there was no change in the rate of entries into the ROI in stimulated mice ($F_{(4,15)}$=0.195, p=0.937, ANOVA; *Figure 6E*). The lack of change in rate of entries into the ROI despite the strong increase in occupancy indicates that mice do not approach or avoid the ROI, contrary to what is expected from either appetitive or aversive reinforcement learning. Second, if ROI stimulation induced a place preference, the effects should persist after pairing (*Liu et al., 2014*). This was also not the case. ROI occupancy and rate of exits assayed on the day immediately after the pairing protocol returned to baseline levels (p>0.05, post-hoc t-tests, with Bonferroni correction, comparing pre and post day, N = 4). Thus, the mice did not learn to prefer or avoid being in the ROI.

Overall, these results appear to be consistent with the hypothesis that the entire effect of ROI stimulation (*Figure 6*) is due to the same transient slowing effect seen in the standard open field assay in which stimulation is delivered in a spatially random fashion (*Figures 1–2*). Activation of DRN 5-HT neurons appears to promote ROI occupancy due to a decrease in movement speed within the ROI, resulting in a decrease in the rate of ROI exits. Interestingly, however, there appeared to be an increase in the effect of stimulation across days, as indicated by the fact that ROI occupancy increase did not reach significance before the third day of stimulation. In addition, there was a decrease in speed within the ROI between the pre- and post-stimulation sessions (p<0.05, post-hoc t-test, *Figure 6G*) that was not seen in WT mice ($F_{(4,20)}$=0.183, p=0.945, 1-way ANOVA across days). Together, these differences suggested the possibility that in addition to transient effects of phasic stimulation there might be longer-term, stimulation-dependent accumulation of effects in SERT-Cre animals.

## Long-term effects of DRN 5-HT photostimulation

In order to test for possible long-term effects of optogenetic activation of DRN 5-HT neurons and to characterize the dynamics of stimulation effects over this longer period, two sets of SERT-Cre and WT mice were tested in the open field arena for >3 weeks. Group 1 (G1, 3 SERT-Cre and 2 WT mice) received DRN 5-HT photostimulation for 24 consecutive days (*Figure 7A–B*). Group 2 (G2, 3 SERT-Cre and 2 WT mice) were exposed to the arena for 23 days with no stimulation; for this group stimulation commenced on the 24th day and lasted for six consecutive days (*Figure 7B*).

Unexpectedly, we observed that the stimulated group (G1) showed a progressive increase in movement speed over the course of 24 days, (*Figure 7C*) while the non-stimulated group (G2) maintained a constant average speed over the same period (*Figure 7E*). This long-term effect was, after 24 days, comparable in magnitude to the transient decreased evoked by stimulation, and manifested both during stimulation and non-stimulation epochs. The average speed increased in G1 from 3.50 ± 0.29 cm·s$^{-1}$ (day 1) to 7.11 ± 0.21 cm·s$^{-1}$ (day 24) during stimulation and 6.28 ± 0.74 cm·s$^{-1}$ (day 1) to 9.03 ± 0.36 cm·s$^{-1}$ (day 24) during control period. These effects were confirmed with a

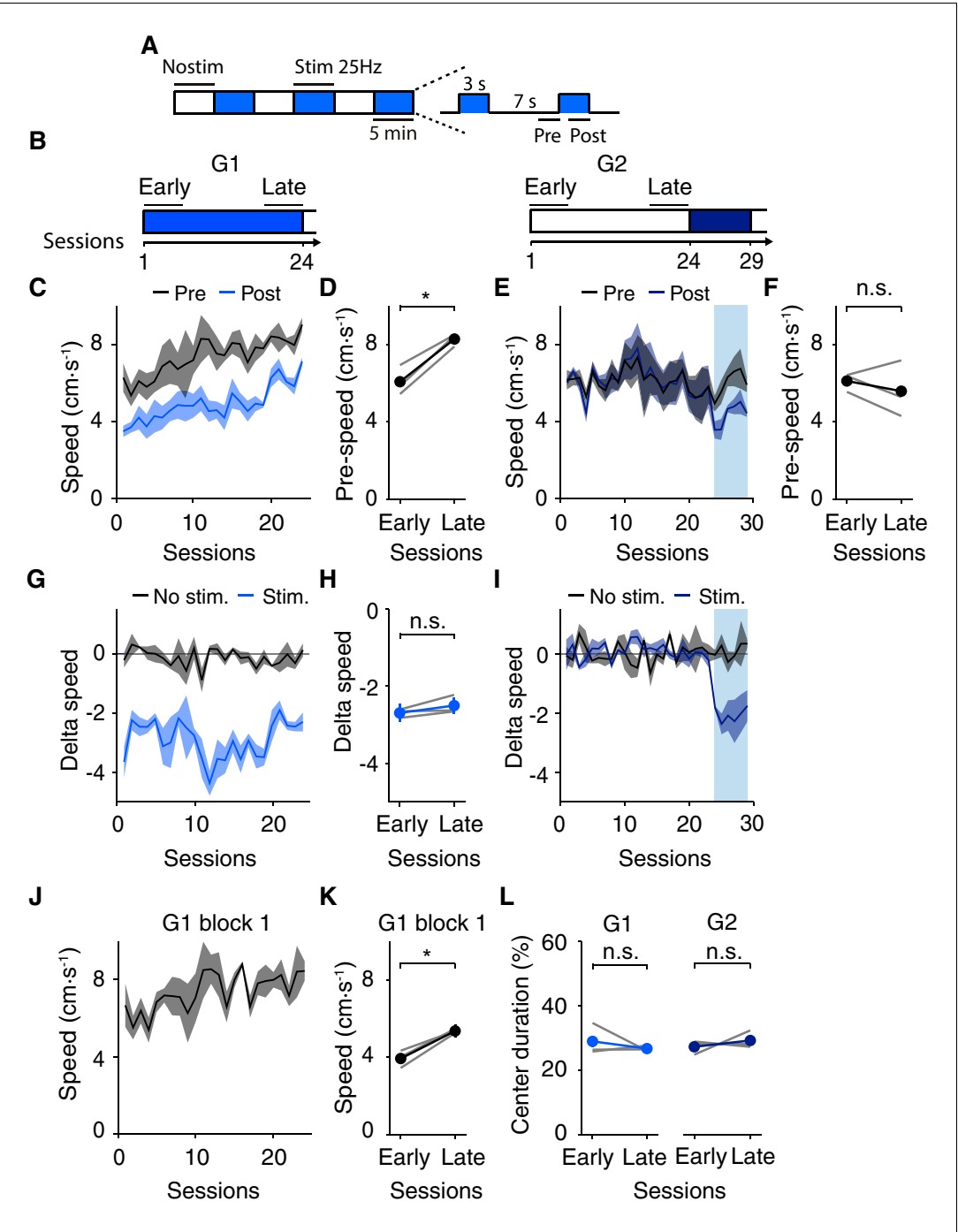

**Figure 7.** Long-term optogenetic DRN 5-HT activation induces an increase in speed in the open field. (**A**) Schematic diagram of the photostimulation protocol, as in *Figure 1D*; a total of 270 s of 20 Hz stimulation is delivered over a 30 min session. (**B**) Experimental protocol. Group 1 (G1, 3 SERT-Cre and 2 WT mice) received photostimulation from session 1 to 24. Group 2 (G2, 3 SERT-Cre and 2 WT mice) was exposed to the arena for 23 days, receiving photostimulation beginning only on the 24th day for six consecutive days. (**C**) Average speed across sessions for Group 1 (N = 3 SERT-Cre mice). Non-stimulated (pre, black) and stimulated (post, blue) intervals. Lines and shading indicate mean ± SEM here and **E**, **G**, **I**, **J**. (**D**) Speed in early and late sessions (as indicated in (**B**)) for mice in (**C**). Dark lines and error bars indicate (mean ± SEM). Error bars are too small to see in some cases. Gray lines indicate individual mice. *p<0.05; n.s: not significant, with paired t test. Same applies to **F**, **H**, **K**, **L**. (**E**, **F**) Same as (**C**, **D**) but for Group 2 (N = 3 SERT-Cre mice). (**G**) Group one stimulation effect (delta speed, difference between post- and pre-stimulation intervals) across all sessions. (**H**) Stimulation effect in early vs.

*Figure 7 continued on next page*

*Figure 7 continued*

late sessions. (I) Same as (G) but for Group 2. (J) Group one speed across sessions assessed during the pre interval only in the first block, i.e. prior to receiving any stimulation during that session (K) Speed in the first block as in (I) for early vs. late sessions (as indicated in (B)). (L) Center occupancy in early vs. late sessions for Group one and Group 2.

The following figure supplement is available for figure 7:

**Figure supplement 1.** Lack of correlation between short and long-term effects of DRN 5-HT activation.

paired t-test, comparing pre-stim. speed in early (1–6) vs. late (19–24) sessions (p=0.00210 for G1, SERT-Cre mice, N = 3, *Figure 7D* and p=0.736 for G2, SERT-Cre mice, N = 3, *Figure 7F*) and by the difference in speed in the non-stimulated epochs between G1 and G2 mice (p=0.0192, paired t-test comparing pre-stim. speed in late sessions, SERT-Cre mice, N = 3, *Figure 7D,F*). The results were further corroborated using generalized linear regression analysis (see Materials and methods). We found that the slope of the regression, which corresponds to a long-term change in speed, was significantly higher than zero only in the mice that were photostimulated repeatedly throughout the experiment (G1; fitted coefficient for session, −0.03, p=0.180; intercept, 6.5, p=$6.51 \times 10^{-42}$; group −0.59, p=0.208; session X group interaction term, 0.15, p=$1.43 \times 10^{-5}$).

Long-term effects were not likely due to changes in effectiveness of stimulation, such as an increase in expression levels, since the magnitude of the transient effect was unaltered over the course of the experiment for G1 (delta speed in early vs. late sessions: −2.69 ± 0.23 vs. −2.50 ± 0.22 cm·s⁻¹ , p=0.212, paired t-test, SERT-Cre mice, N = 3, *Figure 7G–H*). Moreover, even after the induction of the increase in speed in G1, there was no difference in the magnitude of the transient decrease between G2 and G1 (*Figure 7C,E,I*; difference in speed in pre- vs. post-stimulation epochs in the initial sessions of stimulation: 2-way ANOVA with groups and sessions as main factors, sessions, $F_{(5,5)}$=0.317, p=0.884; group, $F_{(1,5)}$=4.10, p=0.0991). Overall, the transient and long-term effects appear to interact roughly additively.

These results indicate that repeated phasic DRN 5-HT optogenetic activation induces two independent and opposite effects: a transient slowing and a sustained speeding of movement. We infer that the sustained effect is a long-term consequence of daily stimulation. However, we considered that the increase in speed might depend on the presence of a period of stimulation on a given day, i.e., that it reflects a 'rebound' from stimulation. To test this we repeated the analysis using only the first 5 min block of each session. This block was never stimulated and any effects in this period could not be due to the effects of recent stimulation, but must rather reflect longer-term effects, at least carried over from the previous day. Indeed, we found a significant increase in speed even during the first non-stimulated block (early vs. late sessions, p=0.0252, paired t-test, SERT-Cre mice, N = 3, *Figure 7J–K*). This suggests that the effect is present from the initial exposure of the animal to the area on a given session and does not depend on immediately preceding stimulation.

Next, we considered whether the long-term effects are somehow related to an anxiolytic effects, as speeding of movement could result from an increase in center excursions, since those tend to be accompanied by a higher velocity (*Figure 5*). We found no increase in center occupancy after long-term stimulation (early vs. late sessions, G1, p=0.201, paired t-test, SERT-Cre mice, N = 3; non-stimulated control G2, p=0.384, paired t-test, SERT-Cre mice, N = 3, *Figure 7L*). Therefore, the sustained effects produced by 15 min per day of phasic DRN 5-HT activation, as the transient effects, do not appear to be secondary to anxiety-related effects measured by thigmotaxis.

Finally, we tested whether session-by-session changes in the magnitude of the long-term effect were predicted by corresponding fluctuations in the short-term effect. However, our analysis failed to reveal such correlations (see Materials and methods and *Figure 7—figure supplement 1*), suggesting that the interaction between the two effects, to the extent that it exists, may require more sensitive behavioral assays to be resolved.

## Discussion

In the present study, we found that optogenetic activation of DRN 5-HT neurons produced two largely independent effects, one short-term and one long-term. The transient effect was a robust decrease in speed in the open field. It was strongly modulated by context and was not accompanied by anxiety-like behaviors or motoric impairments. The long-term effect was an increase in speed in the open field, opposite of the short-term effect, which emerged over the course of three weeks of daily stimulation.

### DRN 5-HT activation suppresses locomotion in a context-dependent manner

The main immediate effect of DRN 5-HT activation was to suppress locomotion. This effect was manifested first as a bias in action selection, with time locomoting and rearing reduced and time stationary enhanced (*Figure 1*). Non-locomotor movements such as grooming and scratching were unaffected and freezing was never observed. DRN 5-HT activation also decreased average speed during locomotion with the magnitude of the decrease positively correlated with the initial movement speed (*Figure 2*). These effects had rapid onset (time constant of approximately 1 s) and offset (3 s). Additional experiments showed that they were also strikingly context dependent. First, there was no effect on performance in the accelerating rotorod, a task that requires a kind of locomotor behavior (*Figure 3*). Second, there was no effect on mice crossing a brightly-lit linear track to obtain water rewards at either end (*Figure 4*). Therefore, suppression of locomotion does not appear to be a direct effect on the motor system, but must instead reflect modulation of factors that drive locomotion.

The effects we observed recall long-standing observations that serotonin changes the effectiveness with which rewards and punishments drive behavior, enhancing the ability of organisms to tolerate delays before acting on such motivations (*Soubrié, 1986*). Such 'behavioral inhibition' has been studied extensively in tasks in which animals must refrain from responding for a delayed reward, including the five choice serial reaction time task (*Harrison et al., 1997*; *Winstanley et al., 2004b*), go/no-go tasks (*Harrison et al., 1999*) and waiting tasks (*Miyazaki et al., 2014*; *Fonseca et al., 2015*). The effects of 5-HT appear to be largest when forces for acting and refraining from acting are balanced (*Soubrié, 1986*). In the open field, there are no overt rewards or punishments motivating behavior. One could infer from the dramatic effects of 5-HT in this context that a normally balanced set of covert factors is shifted toward inaction by 5-HT. Conversely, the lack of effect in linear track or rotorod could reflect the inability of 5-HT to overcome strong local drives present in these contexts (obtaining water rewards and avoiding exposure to bright light in the linear track; avoiding falling in the rotorod).

In the above interpretation, 5-HT would be affecting the manner in which motivating factors such as goals drive actions. Interestingly, goal-directed actions become habitual with extended practice, after which they are no longer driven by the goals, but shift to being stimulus responses, with a corresponding change in neural substrates (*Yin et al., 2004*, *2005*). While 5-HT affects reward waiting times it does not affect stimulus reaction times (*Fonseca et al., 2015*). Therefore, while 5-HT appears to modulate goal-directed actions (*Bailey et al., 2016*) it might not affect habitual actions. Although this hypothesis remains speculative, it would be consistent with the observation of DRN 5-HT activation strongly affected behavior in the open field, in which stimuli-response habits are absent, but failed to affect behavior in the rotorod and linear track, both of which might have become stimulus-response habits. Further experiments would be required to test this.

The selectivity of DRN 5-HT activation effects for certain types of behavior within the open field can also be interpreted in a similar fashion. Whereas locomotion and rearing are considered 'voluntary' (or type I) behaviors, grooming and scratching are considered 'involuntary' (or type II) behaviors (*Vanderwolf, 1988*). 5-HT has been linked to electroencephalographic recordings associated with type I but not type II behaviors (*Vanderwolf and Baker, 1986*; *Vanderwolf, 1989*). This is consistent with our observation that type I behaviors (i.e. locomotion) are modulated by DRN stimulation, whereas type II behaviors (e.g. grooming) seem to be unaffected by it.

## DRN 5-HT activation affects open field behavior independent of anxiety

Our data is consistent with the idea that 5-HT suppresses locomotor activity by decreasing underlying factors that promote it. These data do not reveal exactly how 5-HT achieves this, but do offer some constraints. 5-HT has been classically associated with anxiety and the thigmotaxis (center avoidance) in the open field has been widely used to assay anxiolytic and anxiogenic effects (reviewed in *Prut and Belzung, 2003*). 5-HT could increase the impact of threats that favor immobility over exploration. However, we observed no effects on thigmotaxis (center avoidance) (*Figure 5*), indicating that movement speed effects are not consequences of effects of 5-HT on anxiety but rather direct and independent effects. Our findings are consistent with a recent study showing optogenetic activation of DRN 5-HT neurons decreases locomotor activity without affecting anxiety in the elevated plus maze test (*Ohmura et al., 2014*). Another recent study observed a transient increase in speed in the open field with DRN stimulation (*Warden et al., 2012*). However in that study, both 5-HT and GABAergic neurons in the DRN were targeted, raising the possibility that the protocol was effectively inhibiting DRN 5-HT neurons by driving GABAergic neurons. Two recent studies using pharmacogenetic methods reported an increase in thigmotaxis in the open field following activation of 5-HT neurons (*Teissier et al., 2015*; *Urban et al., 2016*), and one of them (*Teissier et al., 2015*) also reported an overall reduction in locomotion. The additional effects observed could be due to the much slower kinetics of pharmacogenetic methods, which could result in a very different profile of 5-HT transmitter release. The anxiolytic effects of 5-HT appear to be mediated largely by 5-HT1a receptors. It is possible that Ohmura et al. and the present study did not observe these effects because they are not effectively activated by the optogenetic stimulation protocols employed. The 5-HT2a and 5-HT2c receptors have been associated with the effects of 5-HT on impulsivity (*Winstanley et al., 2004a*) and would be candidates to underlie slowing in the open field effects as well.

We found that DRN 5-HT optogenetic activation does not induce anxiety-like behaviors in the open field. However, we do not exclude the possibility that specific sub-populations of 5-HT neurons may modulate anxiety-like behaviors. *Marcinkiewcz et al. (2016)* showed that optogenetic activation of BNST-projecting DRN 5-HT neurons increase anxiety-like behaviors without affecting locomotion. They did not test the effects of overall DRN 5-HT stimulation. The pathway-specific effects might therefore represent an 'off-target' effect of this pathway (*Otchy et al., 2015*) that is not present when a larger portion of the DRN is stimulated, because, it is balanced by an opposite effect through a different DRN output pathway. More experiments are also necessary to identify pathway (s) involved in the DRN 5-HT reduction in locomotion.

## DRN 5-HT activation does not mediate appetitive or aversive reinforcement

Activation of DRN neurons has in some reports been associated with reward or appetitive reinforcement (*Liu et al., 2014*), while other reports have failed to see reinforcing effects (*Miyazaki et al., 2014*; *Fonseca et al., 2015*); reviewed in *Luo et al., 2015*). Here, we revisited this issue using a task nearly identical to that reported previously (*Liu et al., 2014*), delivering DRN 5-HT stimulation repeatedly continent on the location of the animal within a specific ROI (*Figure 6*). We did observe an increase in the occupancy during stimulated sessions, but this could be accounted by the decrease in speed within the ROI, consistent with inhibition of locomotion. Slowing was in turn associated with a decrease in rate of exits from the ROI. In contrast, we observed no change in the rate of entries into the ROI, i.e., mice showed no evidence of increased tendency to approach or avoid the ROI after pairing. Furthermore, ROI occupancy returned to baseline when stimulation ceased. Thus, mice showed no evidence of learning about the ROI. The increase in ROI occupancy could thus be accounted for solely by direct locomotor effects.

It is not clear whether our observations are entirely consistent with those of the previous report, in which ROI occupancy was reported to remain high after cessation of stimulation and entry and exit rates were not reported (*Liu et al., 2014*). Differences between the two sets of results might instead be explained by differences in the populations of DRN neurons stimulated. Whereas we used a SERT-Cre line (*Gong et al., 2007*) to drive ChR2 expression in 5-HT neurons, Liu et al. used the ePet1-Cre line (*Scott et al., 2005*). The Pet-1 line has a broader expression profile, including a 5-HT-negative, glutamate-positive population (*Liu et al., 2014*; *Luo et al., 2015*). DRN reinforcing

effects appear to depend on such a glutamatergic input to the VTA (*Liu et al., 2014*; *McDevitt et al., 2014*; *Qi et al., 2014*) which appears to be absent in the more selective SERT-Cre line (*Fonseca et al., 2015*).

Taking our study and other open field results together with experiments using operant training (*Fonseca et al., 2015*), we believe the evidence currently favors the view that activation of the DRN 5-HT system is not reinforcing. Thus, 5-HT may mirror the DA system in having an opposite direct effect on behavior (devigorating vs. invigorating), but it appears to have neither the same (*Liu et al., 2014*) nor opposite (*Daw et al., 2002*) learning effects.

## Endogenous signaling of 5-HT neurons in relationship to behavioral inhibition

The above observations suggest that activating DRN 5-HT neurons does not affect locomotion by producing a reward- or anxiety-like signal. There remain a variety of possible underlying signals that might be reported by DRN 5-HT neurons and that could result in inhibition of locomotion. Broadly, these could be divided into three domains. A first set of proposals, developed largely in the domain of reinforcement learning theory, posits 5-HT to affect action selection and response vigor by biasing the value of appetitive or aversive stimuli (*Dayan and Huys, 2009*; *Cools et al., 2011*). A second set of ideas posits a signal that modulates the cost of acting (or not-acting) itself (*Miyazaki et al., 2014*; *Fonseca et al., 2015*; *Meyniel et al., 2016*). A third set of ideas is related to the signaling of uncertainty, either in predicting events or the outcomes of ones own actions (*Maswood et al., 1998*; *Amat et al., 2005*; *Matias et al., 2016*). Uncertainty-related signals may be more complex to interpret as they could bias benefits and costs toward either action or inaction depending on the shape of the value landscape or could shift the balance between different behavioral control systems (*Daw et al., 2005*).

There are a number of complexities in relating the firing patterns of DRN 5-HT neurons to the present results. First, other neuromodulators, particularly dopamine and norepinephrine, are also implicated in the same functions (*Yu and Dayan, 2005*; *Dayan and Yu, 2006*) and 5-HT has a variety of features that suggest it acts as an opponent to dopamine (*Daw et al., 2002*; *Cools et al., 2011*; *Boureau and Dayan, 2011*). Our data are consistent with opponent effects of 5-HT and DA as modulators of action vigor, but not as reinforcement learning signals. Simultaneous or parallel recordings from multiple neuromodulator systems will be important for resolving the interplay between these systems (*Cohen et al., 2012*, *2015*; *Matias et al., 2016*). An additional complexity is that, due to differences in receptor kinetics, phasic firing and tonic firing of 5-HT neurons may convey different signals (*Daw et al., 2002*; *Cools et al., 2011*; *Cohen et al., 2015*). Because of the synchronous nature of optogenetic stimulation and the rapid onset of the transient effects, we presume that our protocol mimicked primarily phasic signaling, but we cannot exclude the possibility that it also mimics tonic signals as well. This issue may underlie some of the discrepancies between different pharmacological and optogenetic results. Finally, recordings from DRN 5-HT neurons have indicated substantial diversity in firing patterns (*Nakamura et al., 2008*; *Ranade and Mainen, 2009*; *Cohen et al., 2015*). This might reflect more heterogeneous functions within the DRN that are not discriminated by global DRN stimulation.

## Repeated daily DRN 5-HT activation induces long-term effects

We found that repeated daily exposure to optogenetic DRN 5-HT stimulation produced an unanticipated long-term effect, progressively and persistently increasing locomotion over three weeks (*Figure 7*). These long-term effects were not observed in control animals that were exposed to the same tests but with stimulation withheld until after three weeks. The transient and long-term effects appeared to be opposites and interacted additively. Although further experiments will be needed to address mechanisms, this opponency suggests the possibility of a direct form of compensatory action at a physiological or biochemical level. If the effects of transient DRN 5-HT activation are interpreted as suppression of the influence of motivating factors on locomotion, as suggested above, the long-term effects could represent an enhanced influence of the same motivating factors. The ability of goals to motivate behavior is critical for healthy mental function, and its absence, apathy, is a clinically important factor in many psychiatric disorders (*Chase, 2011*). Although our stimulation protocol was artificial, it was carried out at a rate within the physiological range of 5-HT neurons

(*Nakamura et al., 2008*; *Ranade and Mainen, 2009*; *Cohen et al., 2015*) and would likely produce only a modest increase in overall DRN 5-HT activity. For example, if each optogenetic pulse caused a spike, a neuron with a 1 Hz average firing rate would experience <10% increase in total spikes over a 24 hr period. Therefore, the long-term effects might be ethologically or therapeutically relevant and long-term DRN 5-HT stimulation in the open field could be an interesting paradigm to explore in the context of animals models of affective disorders (*Markou et al., 2013*).

Because the persistent effects of DRN 5-HT activation accumulated gradually over many days and persisted in the absence of stimulation, we interpret them as a form of long-term plasticity. 5-HT has been described to boost plasticity in developing sensory cortices (*Gu and Singer, 1995*; *Maya Vetencourt et al., 2008*, *2011*; *Jitsuki et al., 2011*), possibly modulating the induction of LTP and LTD (*Kojic et al., 1997*; *He et al., 2015*). Notably, SSRIs must be administered for several weeks before anti-depressant effects can be seen, suggesting that anti-depressant effects are not a direct and immediate consequence of changes in 5-HT function but the consequence of an adaptation requiring induction of plasticity (*Branchi, 2011*) or neurogenesis (*Miller and Hen, 2015*). The molecular and temporal specificity afforded by optogenetic access will facilitate the determination of the mechanism of action of the (short and) long-term effects of DRN 5-HT activation. For example, more refined targeting strategies, e.g. retrograde infection (*Rothermel et al., 2013*) or intersectional genetics (*Jensen et al., 2008*), will allow the determination of the pathways involved. This may provide a new window on 5-HT-dependent plasticity.

## Materials and methods

### Animal subjects

Thirty-one adult C57BL/6 mice (19 SERT-Cre mice and 12 wild-type (WT) littermates) were used in this study. A subset of these was used in the long-term open field experiments (6 SERT-Cre, 4 WT), in the accelerating rotarod (7 SERT-Cre, 5 WT) and in the LocoMouse assay (7 SERT-Cre). All procedures were reviewed and performed in accordance with the Champalimaud Centre for the Unknown Ethics Committee guidelines, and approved by the Portuguese Veterinary General Board (*Direcção-Geral de Veterinária*, approval 0421/000/000/2016). The SERT protein is encoded by the Slc6a4 gene and the SERT-Cre mouse line (*Gong et al., 2007*) was obtained from the Mutant Mouse Regional Resource Centers (stock number: 017260-UCD). Male and female mice (18–26 g) were group-housed prior to surgery and individually housed post-surgery and kept under a normal 12 hr light/dark cycle (tested at light phase). Mice had free access to food and water, except seven mice used in the open field test (4 SERT and 3 WT) and the seven SERT-Cre mice used in the LocoMouse assay were under a mild water deprivation protocol. In the LocoMouse experiment, water availability was restricted to the behavioral sessions. Extra water was provided if needed to ensure that mice maintained no less than 85% of their original weight. The open field experiments were run in separate batches using the following order: 1 SERT-Cre, 4 SERT-Cre and 2 WT, 4 SERT-Cre and 3 WT, 3 SERT-Cre and 2 WT, 3 SERT-Cre and 2 WT. The rotarod and LocoMouse experiments were run in two batches of mice: 3 SERT-Cre and 2 WT, followed by 4 SERT-Cre and 3 WT.

### Stereotaxic adeno-associated virus injection and cannula implantation

The surgery procedure is described in more detail at Bio-protocol (*Correia et al., 2017*). For virus injection and cannula implantation, mice were first anesthetized with isoflurane (4% induction and 0.5–1% for maintenance) and placed in a stereotaxic frame (David Kopf Instruments, Tujunga, CA). Lidocaine (2%) was injected subcutaneously before incising the scalp. The skull was covered with a layer of Super Bond C and B (Morita, Kyoto, Japan) to help stabilization of the implant. A craniotomy was drilled over lobule 4/5 of the cerebellum and a pipette filled with a viral solution (AAV2.9.EF1a. DIO.hChR2(H134R)-eYFP.WPRE.hGH, $10^{13}$ GC/mL, University of Pennsylvania) was lowered to the DRN (Bregma −4.7 AP, −2.9 DV) with a 32–33° angle toward the back of the animal. The viral solution (1 μL) was injected using a Picospritzer II (Parker) or an hydraulic pump (UMP3-1, World Precision Instruments, Sarasota, FL), connected to a 5 μl Hamilton syringe (Hamilton, Reno, NV), at a rate of 0.05–0.1 μL/min. An optical fiber (200 μm core diameter, 0.48 NA, 4–5 mm) housed inside a connectorized implant (M3, Doric lenses, Quebec, Canada) was lowered into the brain, through the same craniotomy as the viral injection, and positioned 200 μm above the injection point. The implant

was cemented to the skull using dental acrylic (Pi-Ku-Plast HP 36, Bredent, Senden, Germany). The skin was stitched at the front and rear of the implant. Mice were monitored until recovery from the surgery and returned to their home cage. Gentamicin (48760, Sigma-Aldrich, St. Louis, MO) was topically applied around the implant. Behavioral testing started at least two weeks after virus injection to ensure good levels of expression. Previous studies using the same method reported that 94% of ChR2-YFP positive neurons were serotonergic, assessed with tryptophan hydroxylase immunohistochemistry (*Dugué et al., 2014*).

## Optogenetic stimulation

Light from a 473 nm laser (LRS-0473-PFF-00800–03, Laserglow Technologies, Toronto, Canada or DHOM-M-473–200, UltraLasers, Inc., Newmarket, Canada) was controlled by an acousto-optical modulator (AOM; MTS110-A1-VIS or MTS110-A3- VIS, AA optoelectronic, Orsay, France), except for the LocoMouse assay (controlled directly with custom written software using LabView). The AOM controlled the laser power without any auditory noise and it was triggered by the behavioral control system (Bcontrol), developed by Carlos Brody (Princeton University) in collaboration with Calin Culianu, Tony Zador (Cold Spring Harbor Laboratory, Cold Spring Harbor, NY) and Z.F.M. Light exiting the AOM was collected (KT110/M, Thorlabs, Newton, NJ) into an optical fiber patchcord (200 μm, 0.22 NA, Doric lenses), connected to a second fiber patchcord through a rotary joint (FRJ 1 × 1, Doric lenses), then to a chronically implanted optic fiber cannula through an M3 connector (Doric lenses). Laser power was calibrated using a powermeter (PM130D, Thorlabs) before and after each animal session. The optical fiber patchcord was screwed to the M3 implanted connector at the beginning of each experiment.

## Behavioral procedures

### Open field test

#### Apparatus

The open field apparatus was a 50 × 40 × 30 cm (LxWxH) white box (IKEA, Sweden) illuminated with LED white lights (IKEA, Sweden) to ensure uniform illumination (250–350 lux). The floor of the apparatus was covered with fresh corncob bedding. Mice were placed in the center of the open field arena and allowed to freely explore the environment for 30 min. A video camera (Flea3, Point Grey, CA) was placed directly above the open field, and the x and y position, body centroid and orientation of the animal was continuously tracked in real-time with Bonsai software (*Lopes et al., 2015*). An infrared LED placed outside of the apparatus but in the field of view of the camera was used to synchronize the stimulation protocol with the video and tracking data. All tracking, LED and video data were saved for subsequent offline analysis of the behavior. The open field apparatus was cleaned with 50% ethanol and the bedding was changed between sessions.

#### Stimulation protocol

Optical stimulation consisted of a 3 s long trains of 10 ms pulses at a frequency of 25 Hz and 5 mW amplitude. Previous studies described a robust behavioral effect using these parameters (*Dugué et al., 2014*; *Fonseca et al., 2015*). Sessions were divided into 5 min blocks, alternating between stimulation and control (no stimulation) and always starting with no stimulation, for a total session length of 30 min. During stimulation blocks the 3 s stimulation was delivered every 10 s. In the sessions in which photostimulation frequency was varied (*Figure 2F*) stimulation frequency was randomly chosen on each trial from three possible values [5, 15 or 25 Hz].

#### Long-term open field experiment

Two groups of mice (G1 and G2) were exposed to the open field test repeatedly for 3–4 weeks. G1 (3 SERT-Cre and 2 WT mice) was tested for 24 successive days and photostimulated on each day. G2 (3 SERT-Cre and 2 WT mice) was exposed to the arena for 30 days, but only received photostimulation from 24th day onwards. Stimulation was performed as described in open field section (see *Figure 1*).

## Accelerating rotarod assay

A rotarod (ENV-575M, Med- Associates, St Albans, VT) was set to accelerate from 4 to 40 r.p.m. over a 5 min time period. Mice (7 SERT-Cre and 5 WT) were trained for two consecutive days, with one daily session consisting of 7 trials separated by 5 min resting periods. Mice were placed on the rotarod and trials were considered to have started when the rod began to turn. Trials ended when mice fell from the rod or after 5 min had elapsed. The optical fiber patchcord was connected to the animal and latency to fall and video was recorded in all sessions. The third session (test day) consisted of 10 trials with stimulation randomly assigned to five trials. The photostimulation protocol was identical to the one used in the open field assay (3 s trains every 10 s, 10 ms pulses, 25 Hz, 5 mW).

## LocoMouse assay

The custom-designed LocoMouse setup used to assess whole body coordination during over-ground locomotion in mice was described previously in *Machado et al. (2015)*. Briefly, the linear track apparatus consisted of a 66.5 × 4.5 × 20 cm (LxWxH) clear glass corridor connected to two boxes with water ports. Animals (3 SERT-Cre) were placed on the right-side box and allowed to freely walk across a glass corridor with a mirror below at 45° angle. A single high-resolution, high-speed camera (Bonito CL-400B, Allied Vision Technologies) captured side and bottom views at 400 fps. Acquisition software was written in Labview and a National Instruments PCIe 1433 was used to record and save the movies in real time. Infrared sensors placed at the beginning and end of the corridor detected initiation and termination of each trial. The automatic triggering system allowed mice to self-initiate trials. Animals were water deprived in order to increase their motivation to run from one side to the other of the corridor. Water was delivered (4 µl) in all trials. Mice performed four sessions of 15–60 trials. An infrared LED was synchronized with the stimulation protocol and detection time was collected. Light stimulation (15 ms pulses, 25 Hz, 5 mW) happened in 50% of the trials (randomly assigned) and lasted the time it took the animal to cross the track. All data for the animal tracking, video (paws, nose and tail), LED and sensor detection was collected using the LocoMouse tracking software (https://github.com/careylab/LocoMouse) and saved for subsequent offline analysis of the behavior. The behavioral apparatus was cleaned with 50% ethanol between each animal.

## Real time stimulation conditioned to a region of interest in the open field

The behavioral apparatus was the same as described in the open field test. A region of interest (ROI, 13 × 10.5 cm) was defined using a custom Bonsai workflow (*Lopes et al., 2015*), in order to outline the area of stimulation. No visual cues of the ROI were present in the arena. The experiment consisted of five days, including one baseline session (pre), three sessions with stimulation (ST1-ST3) and a final session without stimulation (post). The optical fiber patchcord was connected to the animal and data for the tracking and video was saved in all sessions. For the stimulation sessions, a Python script was included in the custom Bonsai workflow (*Lopes et al., 2015*) to define the conditioning protocol of stimulation. A digital output signal was sent to a microcontroller board (Uno, Arduino, IT) each time the body centroid of the animal entered in the ROI, producing light stimulation (25 Hz at 10 ms pulses, 20 mW amplitude). The stimulation was immediately terminated when the centroid of the mouse body left the ROI.

We quantified the following parameters for each animal: ROI occupancy (time spent in ROI), rate of entries in ROI (number of entries into the ROI / occupancy time outside the ROI), rate of exits from ROI (number of exits from the ROI / occupancy time inside the ROI), and average speed in the ROI.

All behavioral data is available from the Dryad Digital Repository (*Correia et al., 2016*).

## Histology

Histological analysis were performed after photostimulation experiments to confirm viral expression of ChR2-YFP and optical fiber placement. Mice were deeply anesthetized with pentobarbital (Eutasil, CEVA Sante Animale, Libourne, France) and perfused transcardially with 4% paraformaldehyde (P6148, Sigma-Aldrich). The brain was removed from the skull, stored in 4% paraformaldehyde overnight and kept in cryoprotectant solution (PBS in 30% sucrose) for one week. Sagittal sections (50 µm) were cut in a cryostat (CM3050S, Leica, Germany), mounted on glass slides with mowiol

mounting medium (81381, Sigma-Aldrich, St. Louis, MO). Scanning images for YFP, RFP and transmitted light were acquired with an upright fluorescence microscope (Axio Imager M2, Zeiss, Oberkochen, Germany) equipped with a digital CCD camera (AxioCam MRm, Zeiss) with a 5X or 10X objective.

## Data analysis

All data analysis was performed with custom-written software using MATLAB (Mathworks, Natick, MA). Error bars represent standard error of the mean (SEM).

For the open field and ROI stimulation experiments, the animal's path was recorded by an automated tracking system at 60 fps (Bonsai, *Lopes et al., 2015*). Speed data was smoothed by applying a five-frame median filter. To compare behavior immediately before and after photostimulation we defined 'pre' and 'post' time intervals, respectively. Pre started 2 s before stimulation onset and post 1 s after onset; both bins were 2 s long.

For the ethological characterization of behavior in the open field (*Figure 1E–G*), a trained observer, blind to the experimental condition of the mice, recorded several behavioral states from the video with a custom-written software using Bonsai (*Lopes et al., 2015*). The mobile states included walking (straight locomotor activity), rearing (mouse with both forepaws off the floor) and jumping (mouse with all four paws off the floor). The immobile states included resting (non-locomotor activity that did not include any of the other states), digging (mouse using the forepaws to move the bedding), grooming (rapid cleaning movements of the forepaws towards the face and/or the body) and scratching (very rapid and repeated up and down movements of the hind paws on the side of the body, neck or face).

For the LocoMouse assay, a detailed description of the tracker system can be found in *Machado et al. (2015)*. Briefly, the algorithm's output consisted in 3D trajectories (x,y,z) of the four paws, the snout, and the tail (separated into 15 points) along the trial. All tracks were visually examined and fewer than 10% of trails were removed due to exploratory behaviors. To quantify gait parameters, the trajectories of the individual paws where divided into stride cycles that are composed by swing and stance phase. All strides were sorted into speed bins (5 cm/s bin width) with a minimum of 5 strides per bin, per animal. The bin 15–20 cm/s was selected for presentation, as it included the higher number of trials, for both non-stimulated and stimulated trials.

## Statistical analysis

To compare pre- and post-stimulation speed intervals (*Figure 1E*; *Figure 2C*; *Figure 3D*; *Figure 4C–D,E–G*; *Figure 5B–D,H,K*), no stimulation vs. stimulation trials (*Figure 3D*; *Figure 4C–G*; *Figure 4—figure supplement 1*; *Figure 5E,H,K*) and speed in early vs. late sessions (*Figure 7D,F,H,K,L*) we used a paired t test (ttest, MATLAB). Comparison between independent groups of mice (SERT-Cre vs. WT) was done with a two-sample t test (ttest2, MATLAB). To compare delta speed across quartiles (*Figure 2K*), areas (*Figure 5B–D*) and blocks (*Figure 5E*) we performed a two-way ANOVA (anova2, MATLAB),followed by post-hoc t-tests with Bonferroni correction (multcompare, MATLAB). A two-way ANOVA was further used for comparison of between stimulated and non-stimulated trials across blocks (*Figure 5G,J*) and the pre vs. post speed intervals across sessions (*Figure 7C,E*). To compare between stimulated and non-stimulated trials in the polar plots of the LocoMouse task (*Figure 4—figure supplement 1*) we used a three-way ANOVA applied to a linear mixed model (as described in *Machado et al., 2015*). The results were reported as conditional F tests with Satterthwaite degrees of freedom correction. A one-way ANOVA was used to compare ROI experiment parameters across days (*Figure 6D–G*), followed by post-hoc t-tests with Bonferroni correction (multcompare, MATLAB). For all the ANOVAs performed, when mice were included, they were considered as a random factor. Differences were considered significant at *$p<0.05$, **$p<0.01$ and ***$p<0.001$, unless when Bonferroni correction for multiple comparisons was applied. To compare stim. and non-stim. distributions (*Figure 5I*), we applied a Kolmogorov-Smirnov test. To assess whether increased photostimulation frequency affected movement speed in a dose-dependent manner we used a linear regression analysis, fitting delta speed as a function of frequency (*Figure 2F*). To test for long-term effects of photostimulation on speed (*Figure 7*) we considered the speed during a time window just before photostimulation (pre-stimulation, as defined previously). We regressed these values against session number and group identity according to the following equation:

$$Speed = \beta_0 + \beta_1 \cdot Session + \beta_2 \cdot Group + \beta_3 \cdot Session \cdot Group$$

Where Session stands for the session numberand Group is equal one for G1 mice and 0 for G2 mice.

To analyze the correlation between short- and long-term effects of photostimulation on movement speed and considering that our sample size does not permit analysis at the population level, we analyzed the data within animals and across sessions (for a total of 24 × 3 data points). The hypothesis being tested is that the short-term slowing effect on day n ($speed_{post}(n)$ - $speed_{pre}(n)$) would be related to the long-term speeding effect ($speed_{pre}(n + 1)$ - $speed_{pre}(n)$). Here $speed_{pre}$ is the speed just before stimulation onset and therefore represents the baseline, and $speed_{post}$ is the speed during photostimulation, and n and n+1 refer to consecutive sessions. We also note that a direct correlation analysis between the two quantities could be compromised due to the fact that $speed_{pre}(n)$ appears in both terms. To overcome this limitation we performed the following two analyses:

1. If indeed the magnitude of the short-term effect ($speed_{post,}(n)$ - $speed_{pre}(n)$) is related to the change in baseline speed on the following day ($speed_{pre}(n+1)$ - $speed_{pre}(n)$) then it should also be related to the difference between the speeds during the following and preceding days ($speed_{pre}(n + 1)$ - $speed_{pre}(n-1)$), without suffering from the above mentioned confound (**Figure 7—figure supplement 1A**).
2. Alternatively, assuming the long-term effect grows linearly with time we first performed linear regression of the long term effect as a function of sessions and then examined the correlation between the short-term effect on a particular day ($speed_{post}(n)$ - $speed_{pre}(n)$) and the residual of the linear regression at the subsequent day ($speed_{pre}(n + 1)$ - predicted $speed_{pre}(n + 1)$; **Figure 7—figure supplement 1B**).

## Acknowledgements

We thank members of the Systems Neuroscience Lab for many helpful discussions; Histology, Vivarium, Glass Wash and Media Preparation, Optical Imaging and Microscopy platforms, Enrica Audero, Margarida Duarte, Ana Nunes and João Cruz for technical assistance; Gonçalo Lopes and Niccolò Bonacchi for help with Bonsai software; Catherine French and Costa Lab for the accelerating rotarod; Gil Costa for support with visual diagrams and Champalimaud Research for the fruitful collaborative environment. This work was supported by *Fundação para a Ciência e Tecnologia* (fellowship SFRH / BD/33277/2007 to PAC; SFRH/BD/51210/2010 to ASM), Human Frontier Science Program (fellowship LT000881/2011L to EL), Howard Hughes Medical Institute (International Early Career Scientist to MRC), European Research Council (Advanced Investigator Grants 250334 and 671251 to ZFM; Starting Investigator Grant 640093 to MRC), and the Champalimaud Foundation (ZFM).

## Additional information

### Funding

| Funder | Grant reference number | Author |
|---|---|---|
| Fundação para a Ciência e Tecnologia | SFRH / BD / 33277 / 2007 | Patrícia A Correia |
| Human Frontier Science Program | LT000881/2011L | Eran Lottem |
| Fundação para a Ciência e Tecnologia | SFRH/BD/51210/2010 | Ana S Machado |
| Howard Hughes Medical Institute | International Early Career Scientist | Megan R Carey |
| European Research Council | Starting Investigator Grant 640093 | Megan R Carey |
| European Research Council | Advanced Investigator Grant 250334 | Zachary F Mainen |
| European Research Council | Advanced Investigator Grant 671251 | Zachary F Mainen |

| | |
|---|---|
| Champalimaud Foundation | Zachary F Mainen |

The funders had no role in study design, data collection and interpretation, or the decision to submit the work for publication.

## Author contributions

PAC, Conceptualization, Data curation, Formal analysis, Validation, Investigation, Visualization, Methodology, Writing—original draft, Writing—review and editing; EL, Conceptualization, Data curation, Software, Formal analysis, Validation, Methodology, Writing—review and editing; DB, Conceptualization, Formal analysis, Investigation, Methodology, Writing—review and editing; ASM, Software, Formal analysis, Visualization, Methodology, Writing—review and editing; MRC, Conceptualization, Resources, Supervision, Validation, Writing—review and editing; ZFM, Conceptualization, Resources, Supervision, Funding acquisition, Visualization, Project administration, Writing—original draft, Writing—review and editing

## Author ORCIDs

Megan R Carey, http://orcid.org/0000-0002-4499-1657
Zachary F Mainen, http://orcid.org/0000-0001-7913-9109

## Ethics

Animal experimentation: All procedures were reviewed and performed in accordance with the Champalimaud Centre for the Unknown Ethics Committee guidelines, and approved by the Portuguese Veterinary General Board (Direccì§aÌƒo-Geral de VeterinaÌ ria, approval 0421/000/000/2016).

## Additional files

### Major datasets

The following dataset was generated:

| Author(s) | Year | Dataset title | Dataset URL | Database, license, and accessibility information |
|---|---|---|---|---|
| Correia PA, Lottem E, Banerjee D, Machado AS, Carey MR, Mainen ZF | 2016 | Data from: Transient inhibition and long-term facilitation of locomotion by phasic optogenetic activation of serotonin neurons | http://dx.doi.org/10.5061/dryad.bn1gf | Available at Dryad Digital Repository under a CC0 Public Domain Dedication |

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
