## [Decision Letter]

Thank you for submitting your article "Transient inhibition and long-term facilitation of locomotion by phasic optogenetic activation of serotonin neurons" for consideration by *eLife*. Your article has been reviewed by three peer reviewers, and the evaluation has been overseen by a Reviewing Editor and a Senior Editor. The following individuals involved in review of your submission have agreed to reveal their identity: Naoshige Uchida (Reviewing Editor and Reviewer #1), Sebastien Bouret (Reviewer #2) and Garret D. Stuber (Reviewer #3).

The reviewers have discussed the reviews with one another and the Reviewing Editor has drafted this decision to help you prepare a revised submission.

Summary:

This study examined the effect of stimulating serotonergic neurons in the dorsal raphe (DR) to decipher functions of DR serotonin neurons. The authors performed a series of experiments to tease apart potential confounding factors and the results are carefully interpreted. The results overall advance our understanding of serotonin functions. Importantly, this study also points out important interpretational problems in a commonly used real-time place preference assay. All the reviewers agreed that this study is very interesting and warrants publication in *eLife*. However, the reviewers raised some important issues that the authors should address before publication.

Essential points:

1) As reviewer 1 points out, some conclusions are based on small numbers of animals. The authors should obtain data from more animals or interpret these results more carefully.

2) As reviewer 2 points out, the authors' conclusion that the effect of stimulation biased action selection, rather than it caused movement inhibition, is not convincing, and the authors should explain this better or reduce the tone of this conclusion.

3) The authors tend to compare effects across conditions by comparing the results of independent t tests. (e.g., subsection “Effect of DRN 5-HT photostimulation does not induce anxiety-like behavior”, second paragraph for speed as a function of spatial area; subsection “Spatially-specific DRN 5-HT optogenetic activation does not yield place preference or avoidance”, third paragraph for speed and ROI, see also subsection “Long-term effects of DRN 5-HT photostimulation”, third paragraph). I am not questioning the conclusions of the authors (that the effects are similar across conditions), but to support that claim, they should test directly for a difference between the size of the effects between the 2 conditions. This could be done (for instance) by looking at the interaction between the 2 factors, with an ANOVA, as they did in the third paragraph of the subsection “Effect of DRN 5-HT photostimulation does not induce anxiety-like behavior”. Or by computing the effect of stimulation on speed and assess the effect of session on that measure using a simple linear regression.

---

## [Author Response]

*Essential points:*

*1) As reviewer 1 points out, some conclusions are based on small numbers of animals. The authors should obtain data from more animals or interpret these results more carefully.*

We performed a new set of experiments, using 4 SERT-Cre and 3 WT mice in the LocoMouse and rotarod tasks, which are included in Figure 3, Figure 4 and Figure 4—figure supplement 1. With a total of 7 SERT-Cre and 5 WT mice, we observed similar results and maintain our claim that DRN 5-HT activation does not induce general motor impairment (please see update in subsection “DRN 5-HT activation does not induce general motor impairment”, second and last paragraphs. As for the long-term open field assay (Figure 7), this experiment would require a longer time scale (6-7 weeks only for the experimental execution), therefore, we hope the editor and reviewers understand that we were not able to obtain more data. However, as suggested by reviewer #2, we performed more analysis to look at the correlation between transient and chronic effects.

2) As reviewer 2 points out, the authors' conclusion that the effect of stimulation biased action selection, rather than it caused movement inhibition, is not convincing, and the authors should explain this better or reduce the tone of this conclusion.

We thank the reviewer for insisting on this point. After considering this carefully, we agree with the reviewer that it is more parsimonious to interpret our results as “in terms of 'decreased motivation' (=decreased driving force for movement)*”,* as it was suggested. We modified our Discussion accordingly by merging and rewriting the first two sections. Please see the new discussion in “DRN 5-HT activation suppresses locomotion in a context-dependent manner” as well as corresponding changes in the Abstract and Introduction (deleted last paragraph).

*3) The authors tend to compare effects across conditions by comparing the results of independent t tests. (e.g., subsection “Effect of DRN 5-HT photostimulation does not induce anxiety-like behavior”, second paragraph for speed as a function of spatial area; subsection “Spatially-specific DRN 5-HT optogenetic activation does not yield place preference or avoidance”, third paragraph for speed and ROI, see also subsection “Long-term effects of DRN 5-HT photostimulation”, third paragraph). I am not questioning the conclusions of the authors (that the effects are similar across conditions), but to support that claim, they should test directly for a difference between the size of the effects between the 2 conditions. This could be done (for instance) by looking at the interaction between the 2 factors, with an ANOVA, as they did in the third paragraph of the subsection “Effect of DRN 5-HT photostimulation does not induce anxiety-like behavior”. Or by computing the effect of stimulation on speed and assess the effect of session on that measure using a simple linear regression.*

We performed more statistical analysis, as suggested by the reviewers. We included a 2-way ANOVA for speed across different areas (Figure 5):

“Nevertheless, DRN 5-HT activation caused a robust decrease in speed regardless of the spatial area where the animal was located at stimulation onset (Figure 5), as indicated by comparing speed between pre and post stimulation intervals across the different areas (2-way ANOVA, stimulation, areas; SERT-Cre mice, N = 15; stimulation, F_(1,84)_ = 250.123, p = 2.58x10^-17^; areas, F_(2,84)_ = 18.867, p = 3.99x10^-4^; stimulation x areas, F_(2,84)_ =8.576, p = 2.38 x 10^-2^).”

And linear regression analysis in the long-term experiment to compare speed, sessions and groups (Figure 7):

“The results were further corroborated using generalized linear regression analysis (see Experimental procedures). We found that the slope of the regression, which corresponds to a long-term change in speed, was significantly higher than zero only in the mice that were photostimulated repeatedly throughout the experiment (G1; fitted coefficient for session, -0.03, p = 0.18; intercept, 6.5, p = 6.51x10^-42^; group -0.59, p = 0.208; session X group interaction term, 0.15, p = 1.43x10^-5^).”

However, a 2-way ANOVA is not applicable in the case of the ROI analysis (Figure 6) since the multiple ANOVAs (Figure 6) are not related to different conditions but rather to different dependent variables (ROI occupancy, nr entries, nr exits and speed).